# Mechanotransduction current is essential for stability of the transducing stereocilia in mammalian auditory hair cells

A Catalina Vélez-Ortega, Mary J Freeman, Artur A Indzhykulian[†], Jonathan M Grossheim, Gregory I Frolenkov*

Department of Physiology, College of Medicine, University of Kentucky, Lexington, United States

**Abstract** Mechanotransducer channels at the tips of sensory stereocilia of inner ear hair cells are gated by the tension of 'tip links' interconnecting stereocilia. To ensure maximal sensitivity, tip links are tensioned at rest, resulting in a continuous influx of $Ca^{2+}$ into the cell. Here, we show that this constitutive $Ca^{2+}$ influx, usually considered as potentially deleterious for hair cells, is in fact essential for stereocilia stability. In the auditory hair cells of young postnatal mice and rats, a reduction in mechanotransducer current, via pharmacological channel blockers or disruption of tip links, leads to stereocilia shape changes and shortening. These effects occur only in stereocilia that harbor mechanotransducer channels, recover upon blocker washout or tip link regeneration and can be replicated by manipulations of extracellular $Ca^{2+}$ or intracellular $Ca^{2+}$ buffering. Thus, our data provide the first experimental evidence for the dynamic control of stereocilia morphology by the mechanotransduction current.

**\*For correspondence:** Gregory. Frolenkov@uky.edu

**Present address:** [†]Department of Neurobiology, Harvard Medical School, Boston, United States

**Competing interests:** The authors declare that no competing interests exist.

## Introduction

The sense of hearing depends on stereocilia, the microvilli-like mechanosensory projections at the apical surface of inner ear hair cells. A hair cell bundle consists of stereocilia rows with precisely graded heights according to the cell's location along the cochlea, suggesting that the exact shape of the bundle is crucial for the normal function of the hair cell (*Engström and Engström, 1978*). Mammalian auditory hair cells do not regenerate and, therefore, have to maintain their precisely arranged stereocilia throughout the lifespan of the organism. Indeed, recent data show that the actin core is remarkably stable along the length of stereocilia, except for a small region at their tips (*Zhang et al., 2012*; *Drummond et al., 2015*; *Narayanan et al., 2015*).

Hair cell stereocilia are interconnected by extracellular 'tip links' (*Pickles et al., 1984*). Sound-induced deflections of a hair bundle modulate the tension of the tip links, which controls the opening of mechano-electrical transduction (MET) channels (*Assad et al., 1991*). These channels are located at the tips of shorter but not the tallest row stereocilia (*Beurg et al., 2009*). In the resting bundle, the tip links are under a certain degree of tension, ensuring responses to the smallest sound-induced deflections (*Howard and Hudspeth, 1987*; *Hacohen et al., 1989*; *Assad et al., 1991*). This resting tension is thought to be responsible for the wedge-shaped stereocilia tips at the lower end of the tip links in shorter row stereocilia (*Furness and Hackney, 1985*; *Kachar et al., 2000*; *Rzadzinska et al., 2004*). In addition, the resting tip link tension increases the open probability of MET channels, resulting in a continuous influx of $Ca^{2+}$ into the cell through these non-selective cation channels (*Corey and Hudspeth, 1979*). This constitutive $Ca^{2+}$ influx is perceived as a potentially deleterious consequence of the extreme sensitivity of the MET apparatus in the auditory hair

**eLife digest** Our sense of hearing depends on cells known as hair cells that line the inner ear. Each hair cell has tiny projections called stereocilia, which are arranged in a bundle with rows of increasing height like a staircase and are connected to each other by tiny filaments called tip-links. When sound waves hit the stereocilia, the tension on the tip-links increases, which opens "mechanotransduction" channels on the shorter stereocilia that allow calcium ions to flow into the cells. To ensure that the ears can detect even the softest sounds, the tip-links always have a small amount of tension which allows a small, but continuous flow of calcium ions into the cell. Scientists generally consider this continuous flow of calcium ions as a potentially harmful byproduct of sensitive hearing.

Vélez-Ortega et al. isolated inner ear tissues from young mice and rats and exposed them to drugs that either block the flow of calcium ions through the mechanotransduction channels or break the tip-links on stereocilia. Surprisingly, these drugs made profound changes in the shape of individual stereocilia and the staircase architecture of the stereocilia bundle. When the drugs were rinsed out of the hair cells, the stereocilia went back to their normal shape. Sequestering of free calcium ions inside the hair cells had a similar effect on the shape of stereocilia. These findings show that the flow of calcium ions into the sterocilia via mechanotransduction channels controls the exquisite staircase-like architecture of the stereocilia bundle.

More research is needed to identify which structural proteins cause the stereocilia shape changes and to work out exactly how calcium ions are involved.

cells (*Beurg et al., 2010*). However, we show here that the resting MET current also controls the structural stability of the transducing stereocilia in auditory hair cells.

## Results

### Blockage of the MET channels leads to length dysregulation and overall shortening of transducing stereocilia

In our previous study (*Indzhykulian et al., 2013*), we noticed relatively slow changes (within ~20 min) of the stereocilia tip shape after breaking the tip links. We observed these changes in scanning electron microscopy (SEM) images obtained from samples fixed at different time points after tip link breakage. Therefore, we decided to explore whether these changes of stereocilia tip shape could be initiated by the loss of the resting MET current. We blocked the MET channels with extracellular amiloride or benzamil at concentrations of 100 µM and 30 µM, respectively. At these concentrations, the blockers are expected to inhibit ~75% (amiloride) or ~90% (benzamil) of the MET current induced by hair bundle deflections (*Rüsch et al., 1994*). Indeed, both these blockers substantially reduced the entry of FM1-43 dye through the partially open at rest MET channels in both inner (IHCs) and outer (OHCs) hair cells of young postnatal mouse organ of Corti explants (*Figure 1A*). We also cultured organ of Corti explants at postnatal day four (P4) in the presence of benzamil for 24 hr and confirmed that, after a long-term MET blockage, the MET channels remain inhibited but are still functional (*Figure 1B*).

Next, we used scanning electron microscopy (SEM) to examine the effect of this long-lasting inhibition of MET channels on the morphology of the hair bundle. Although a small re-shaping of the stereocilia tips was expected based on previous observations after breakage of the tip links (*Kachar et al., 2000*; *Rzadzinska et al., 2004*), we were surprised to discover that both amiloride and benzamil caused dramatic changes to the staircase morphology of OHC stereocilia bundles both in mice (*Figure 1C*) and in rats (*Figure 1—figure supplement 1*). To quantify these changes, we acquired SEM images of the same hair cell bundles from different viewing angles (between 0° and ±52°) in the mid-cochlear region (*Figure 1—figure supplement 2*). Typically, we determined an angle of view to the top of the bundle and obtained at least two views of each side of the bundle (medial and lateral), which ensured accurate measurements of stereocilia heights (*Figure 1—figure supplement 3*). Our analysis showed that neither of the MET channel blockers affected the height of

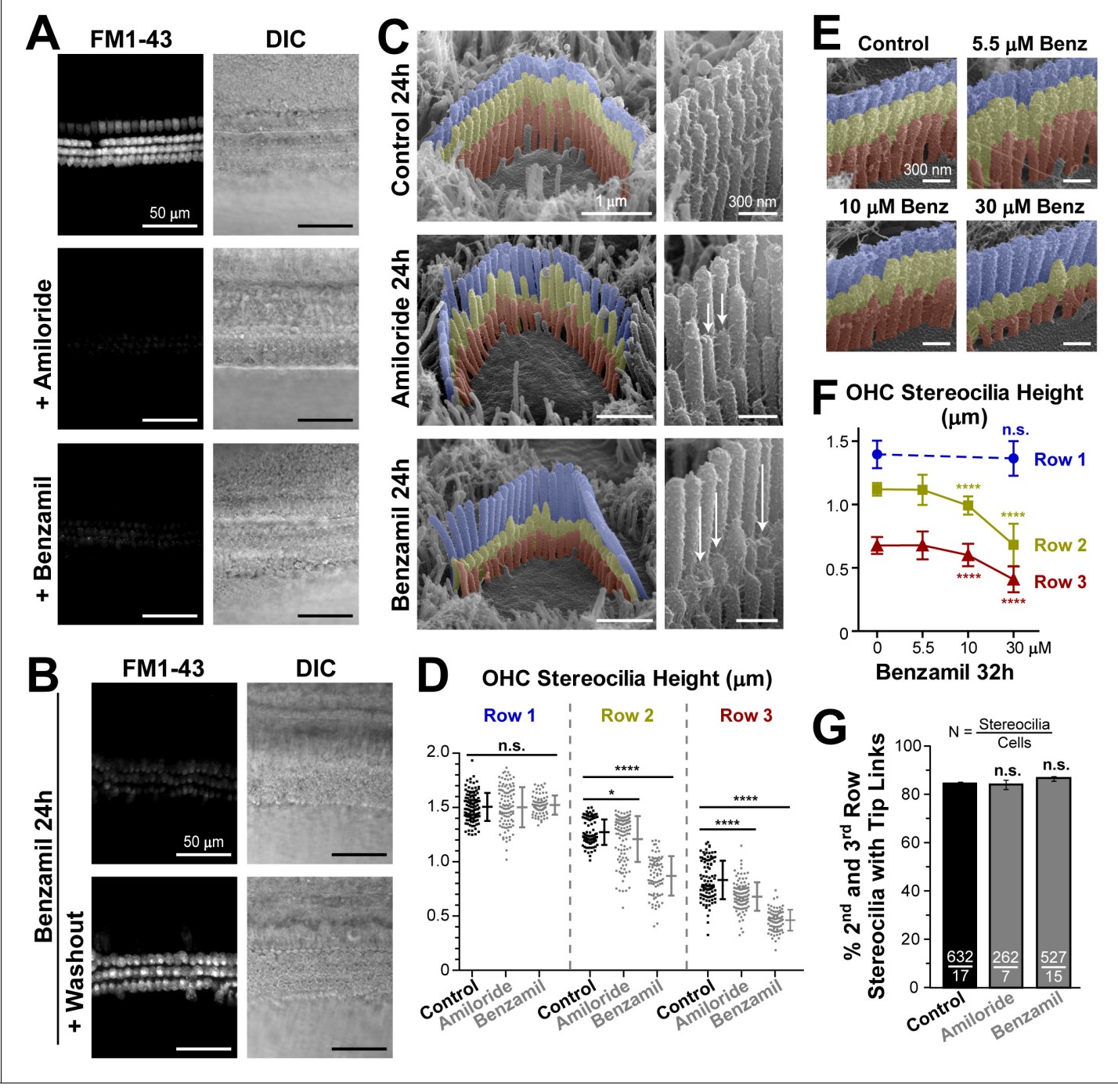

**Figure 1.** Long-term blockage of the MET channels causes selective shortening of the second and third, but not the first (tallest), rows of stereocilia in mouse outer hair cell (OHC) bundles. (**A and B**) Assessment of MET blockage with MET channel-permeable dye, FM1-43. (**A**) Left panels show maximal projection images of FM1-43 fluorescence in mouse organ of Corti explants immediately after the tissue dissection, in control conditions (top) and in the presence of non-saturating concentrations of MET blockers: amiloride (100 µM, middle) or benzamil (30 µM, bottom). Right panels show reference bright-field images of the same cochlear explants at the focal plane of the hair cell bodies. Data are representative of two independent series. (**B**) Similar maximal projection FM1-43 (left) and bright-field (right) images at the end of 24 hr incubation at 37°C with 30 µM of benzamil (top) and after washout of this drug (bottom). (**C**) Representative scanning electron microscopy (SEM) images of OHC stereocilia bundles (false-colored) in mouse organ of Corti explants cultured for 24 hr at 37°C in vehicle control conditions (top), 100 µM of amiloride (middle), or 30 µM of benzamil (bottom). Right panels show higher magnification images of OHC stereocilia. Arrows point to examples of retracted stereocilia. (**D**) Heights of individual stereocilia in different rows of OHC bundles in mouse organ of Corti explants cultured for 24 hr in control conditions (black; n = 103–120 stereocilia) or in the presence of the MET blockers (gray), amiloride (100 µM, n = 99–108) or benzamil (30 µM, n = 75–80). Error bars indicate mean ± SD. The data are from
*Figure 1 continued on next page*

*Figure 1 continued*

a single series of experiments (8–17 cells per treatment) with control and drug-treated explants processed in parallel, representative of one (amiloride) and three (benzamil) independent series. (E and F) Representative false-colored SEM images of OHC bundles (E) and quantification of stereocilia heights (F) in the first (blue), second (yellow) and third (red) rows of the bundle (n = 40–130), indicating the dose-dependent effect of a 32 hr incubation in the presence of 0, 5.5, 10 and 30 μM of benzamil. Data (4–12 cells per treatment) are shown as mean ± SD. For D and F: *p<0.05; ****p<0.0001; n.s., non-significant (Welch's *t* tests). (G) Percentage of shorter (second and third) row OHC stereocilia having tip links after 24 hr culturing in control conditions (black, n = 632) or with MET blockers (gray, n = 262–527). Combined data from two independent series (7–17 cells per treatment) are shown as mean ± SE. n.s., non-significant (Student's *t* tests). Age of explants A: P5; B–G: P4 +24–32 hr incubation. Original SEM images can be found in ***Vélez-Ortega et al. (2017)***.

The following figure supplements are available for figure 1:

**Figure supplement 1.** Blockage of the MET current causes selective shortening of the second- and third-row (transducing) stereocilia in rat auditory hair cells.

**Figure supplement 2.** Organ of Corti examinations were limited to the middle cochlear region.

**Figure supplement 3.** Quantification of stereocilia heights from SEM images.

the tallest (first row) stereocilia in OHC bundles (*Figure 1D*, left). Of note, similar heights of the tallest row stereocilia were reported in hamster OHCs at the same mid-cochlear location and the same age (P5) (*Kaltenbach et al., 1994*). Furthermore, throughout the 24 hr culturing of the mouse cochlear explants harvested at P4, we did not detect any height changes in the tallest row stereocilia in the mid-cochlear OHCs (data not shown), consistent with previous reports showing that OHC (but not IHC) stereocilia stop growing and reach a plateau at the ages of P2-P4 throughout most of the cochlea except at the very apex (*Roth and Bruns, 1992*; *Kaltenbach et al., 1994*).

In contrast to the non-transducing tallest row stereocilia, transducing stereocilia of the second and third rows in the OHC bundles exhibited dramatic changes of their morphology after incubation with MET channel blockers (*Figure 1C*). Especially in response to benzamil, many stereocilia shortened to heights that we never observed in control samples processed in parallel. These results indicate the retraction or disassembly of the stereocilium actin core (*Figure 1C*, bottom and *Figure 1D*, right columns). The effect of amiloride was similar, but it produced larger variability of stereocilia heights (*Figure 1C*, middle and *Figure 1D*, middle columns). Overall, we detected a decrease in the average height of the second- and third-row stereocilia after incubation with either of the MET channel blockers (*Figure 1D*). The effect of MET current blockage on the height of transducing stereocilia was dose-dependent (*Figure 1E,F*). Among the different rows of OHCs in the organ of Corti, the largest effects were observed in the third (the outermost) row of OHCs (data not shown). Notably, the number of visible tip links per stereocilium in OHCs did not change after long-term MET blockage (*Figure 1G*). It is not clear whether these tip links are newly formed links or the same links that slid down. However, their presence suggests that stereocilia shortening after the blocking of the MET channels is likely to occur even in the presence of tip link-generated mechanical tension and other tip link-associated signaling events.

The effects of MET blockers on the staircase morphology of IHC bundles were qualitatively similar to those observed in OHCs (*Figure 2*). However, IHCs exhibited a smaller decrease in the heights of transducing stereocilia as compared to OHCs from the same explants and location along the cochlea (*Figure 2A–B*). As in OHCs, long-term blockage of MET channels in IHCs did not result in the loss of tip links (*Figure 2C*). We also noticed the accelerated 'pruning' of the supernumerary (fourth and fifth rows and unranked) stereocilia after MET channel blockers in both IHCs (control = 25.1 ± 0.9 vs. benzamil = 20.4 ± 1.3 supernumerary stereocilia per cell, n = 19–31 cells, p<0.04) and OHCs (control = 10.9 ± 0.8 vs. benzamil = 6.8 ± 0.7, n = 32–65, p<0.002). It may indicate that these supernumerary stereocilia express functional MET channels or that the developmentally regulated program of their retraction depends on the intracellular $Ca^{2+}$ concentration, which is expected to decrease after MET channel blockage.

Besides amiloride and benzamil, a similar retraction of shorter (but not the tallest) row stereocilia was observed in mouse OHCs and IHCs after 24 hr incubation with 30 μM of tubocurarine (data not

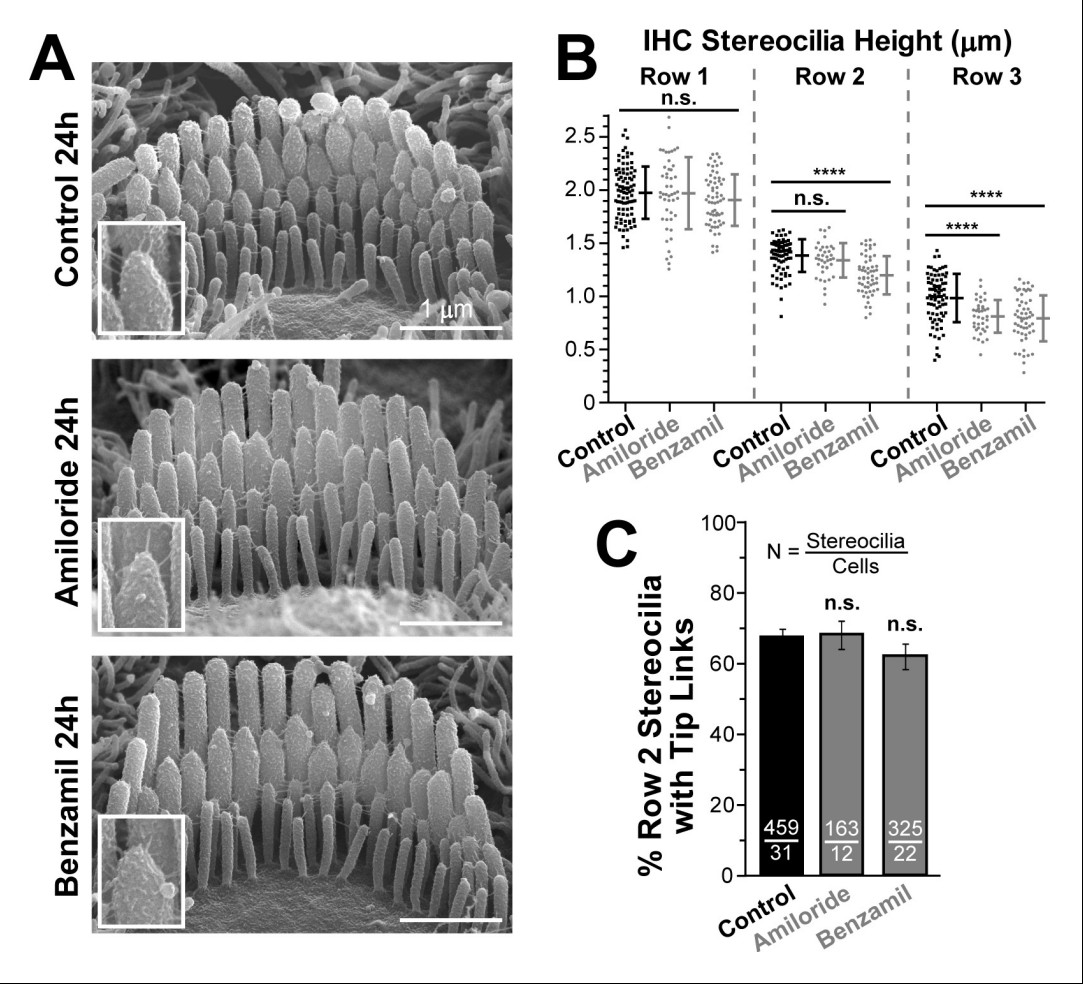

**Figure 2.** Blockage of the MET channels causes selective shortening of transducing second- and third-row stereocilia but not the tallest first-row stereocilia in mouse inner hair cells (IHCs). (**A**) Representative scanning electron microscopy (SEM) images of IHC stereocilia bundles in mouse organ of Corti explants cultured for 24 hr at 37°C in vehicle control conditions (top), 100 µM of amiloride (middle), or 30 µM of benzamil (bottom). The insets show higher magnification images of the tips of second-row stereocilia. (**B**) Heights of individual stereocilia in different rows of IHC bundles in mouse organ of Corti explants cultured for 24 hr in control conditions (black; n = 80–91 stereocilia) or in the presence of the MET blockers (gray), amiloride (n = 38–48) or benzamil (n = 55–62). The heights of the tallest stereocilia are similar to those reported for the hamster IHCs at the same age and mid-cochlear location (*Kaltenbach et al., 1994*). Error bars indicate Mean ± SD. *p<0.05; ****p<0.0001; n.s., non-significant (Welch's *t* tests). The data are from a single series of experiments (8–18 cells per treatment) with control and drug-treated explants processed in parallel, representative of one (amiloride) and three (benzamil) independent series. (**C**) Percentage of the second-row stereocilia with tip links in IHC bundles after 24 hr incubation in control conditions (black, n = 459) or with MET blockers (gray, n = 163–325). The data from two independent series are shown (12–31 cells per treatment) as mean ± SE. n.s., non-significant (Student's *t* tests). Age of explants in **A-C**: P4 +24 hr incubation.

shown), a larger molecule that also blocks MET channels but is unlikely to permeate through them (*Farris et al., 2004*). Altogether, the selective shortening of the transducing second- and third-row stereocilia, but not of those in the tallest row, argues against a non-specific action of the MET blockers on the actin core of the stereocilium.

## MET-dependent stereocilia shortening recovers after washout of the MET blockers

Next, we explored whether the observed stereocilia shortening was permanent. We cultured several organ of Corti explants with or without benzamil for 24 hr and, as expected, found retraction of the transducing stereocilia (*Figure 3A,C*). Then, we removed the benzamil and allowed the explants to recover at 37°C for an additional 24 hr. After the recovery period, the second-row stereocilia and the majority of third-row stereocilia re-grew and reached the same stereocilia heights as in the control explants that were processed in parallel (*Figure 3B,D*). This data suggest that the effects of the MET blockers on the transducing stereocilia are reversible and indicate that the MET current is likely to dynamically regulate the height of these stereocilia.

## MET-dependent shortening of transducing stereocilia begins with thinning at their tips

In the next experiment, we compared the effects of three chemically unrelated MET blockers. We used benzamil, ruthenium red, and tubocurarine, at concentrations of 30, 10 and 30 µM,

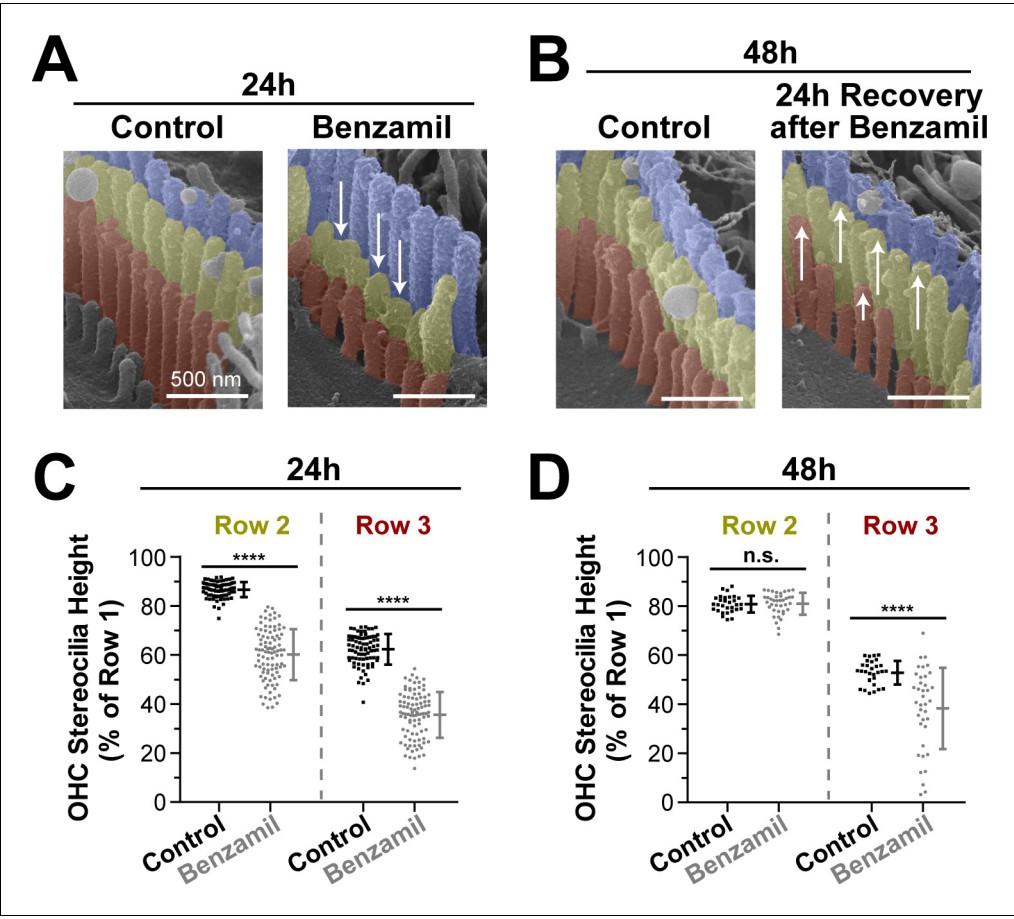

**Figure 3.** Transducing stereocilia that have retracted due to MET blockage are able to regrow after drug washout. (**A and B**) Representative false-colored SEM images of mouse OHC stereocilia after (**A**) 24 hr incubation either in control conditions (left) or with 30 µM of benzamil (right), and after (**B**) 24 additional hours of recovery after washout. Arrows down point to retracted stereocilia, while arrows up indicate re-growth. (**C and D**) Heights of second- and third-row stereocilia after (**C**) 24 hr incubation in control conditions (black, n = 83) or with benzamil (gray, n = 95), and (**D**) 24 hr after washout (n = 30, control; n = 38, benzamil). Stereocilia heights are shown as a percent relative to the size of tallest (first) row. Data are from 7 to 16 cells per treatment. Error bars indicate mean ± SD. ****p<0.0001; n.s., non-significant (Welch's *t* tests). Age of the explants: P4 +24–48 hr. All incubations were performed at 37°C.

respectively, previously shown to block 80–90% of the MET current (*Rüsch et al., 1994*; *Farris et al., 2004*). We used a shorter incubation period, but, as before, we processed all samples in parallel and imaged the hair cell bundles at the same mid-cochlear location. As expected, the incubation with benzamil, ruthenium red or tubocurarine for 5 hr led to a smaller, but still statistically significant, retraction of shorter row stereocilia in the mouse OHCs (*Figure 4A,B*). Although all these chemically unrelated drugs may have different side effects, the fact that they all produced qualitatively similar shortening of transducing stereocilia in the OHCs indicates that such shortening is likely to arise from their common action—the blockage of the MET channels.

The 5-hr incubation with MET blockers also allowed us to explore the initial steps of stereocilia shortening. After this incubation, we observed an increased number of thin and 'pointed' tips in the second- and third-row stereocilia of OHC bundles but not in the non-transducing tallest row stereocilia (*Figure 4A*—arrowheads, and *Figure 4—figure supplement 1*). We quantified these shape changes in the second row of stereocilia, where the effect was most prominent. We found that all MET channel blockers tested produce highly significant thinning of the tips of these stereocilia in OHCs (*Figure 4C*). We examined the actin core of the abnormally thin stereocilia tips with transmission electron microscopy (TEM) in plunge-frozen freeze-substituted preparations. In all these stereocilia (n = 30), actin filaments filled the tips completely, without any signs of 'over-tented' membranes (*Figure 4D*). We concluded that the MET current-dependent thinning of the tips of transducing stereocilia is caused by the remodeling of the stereocilia actin core. In fact, these data also suggest that the actin filaments located at the circumference of the stereocilia core, farther away from the transducer channel, are more susceptible to the blockage of the MET current.

The 5-hr incubation with MET channel blockers affected transducing stereocilia in IHCs to a lesser degree than in OHCs (*Figure 5A*). These differences between IHCs and OHCs were expected based on the results observed after the 24 hr incubations (*Figures 1* and *2*). However, we still detected statistically significant changes in the staircase 'steps' of IHC bundles after the 5-hr incubation with some (but not all) MET channel blockers (*Figure 5B*). Despite the larger variability of tip shapes in the second-row stereocilia in IHCs as compared to OHCs, we were also able to detect statistically significant changes in the shape of IHC second-row stereocilia tips after incubation with any of these three MET channel blockers (*Figure 5C*). Similar to OHCs, we observed the remodeling of the underlying actin core after MET current blockage (*Figure 5D*) in all second-row IHC stereocilia with abnormally thin tips that we examined by TEM (n = 10). We concluded that the effects of MET current blockage on the shapes of IHC stereocilia tips are less prominent but qualitatively similar to those in OHCs.

## Ca$^{2+}$ influx through the MET channels controls the remodeling of transducing stereocilia

To determine whether Ca$^{2+}$ influx is the component of the MET current responsible for maintaining the stability of transducing stereocilia, we loaded organ of Corti explants with the membrane-permeable acetoxymethyl (AM) ester derivative of BAPTA (BAPTA-AM), which is cleaved inside the cell by endogenous esterases. This treatment results in the accumulation of BAPTA in the cell, increasing intracellular Ca$^{2+}$ buffering and limiting any Ca$^{2+}$-dependent effects to the vicinity of the sites of Ca$^{2+}$ entry into the cytosol. It does not affect the integrity of stereocilia links (*Figure 6A,B*, insets), because micromolar concentration of BAPTA-AM outside of the cell is not sufficient to significantly decrease the millimolar concentration of extracellular Ca$^{2+}$. In both OHCs and IHCs, intracellular BAPTA resulted in the appearance of the abnormally thin tips in the second but not the tallest row stereocilia (*Figure 6A–D*) as well as the shortening of many transducing second- and third-row stereocilia (*Figure 6E,F*). Similar to our results with extracellular MET blockers, the effects of intracellular BAPTA on the average height of transducing stereocilia were more prominent in OHCs than in IHCs (*Figure 6E,F*). This stereocilia remodeling after BAPTA-AM cannot be attributed to the loss of MET current because intracellular BAPTA does not block this current or change the resting open probability of MET channels in either OHCs or IHCs, even at a very large concentration of 10 mM (*Peng et al., 2013*). However, the intracellular BAPTA should decrease the size of the Ca$^{2+}$'hotspot' at the point of Ca$^{2+}$ influx through the MET channels (*Figure 6G*). A steeper Ca$^{2+}$ gradient inside the stereocilium would decrease the Ca$^{2+}$ concentration experienced by the peripheral actin filaments, causing their preferential retraction or disassembly (*Figure 6G*, arrows) and resulting in a 'pointed' shape at the tips of second-row stereocilia in both OHCs (*Figure 6C*) and IHCs

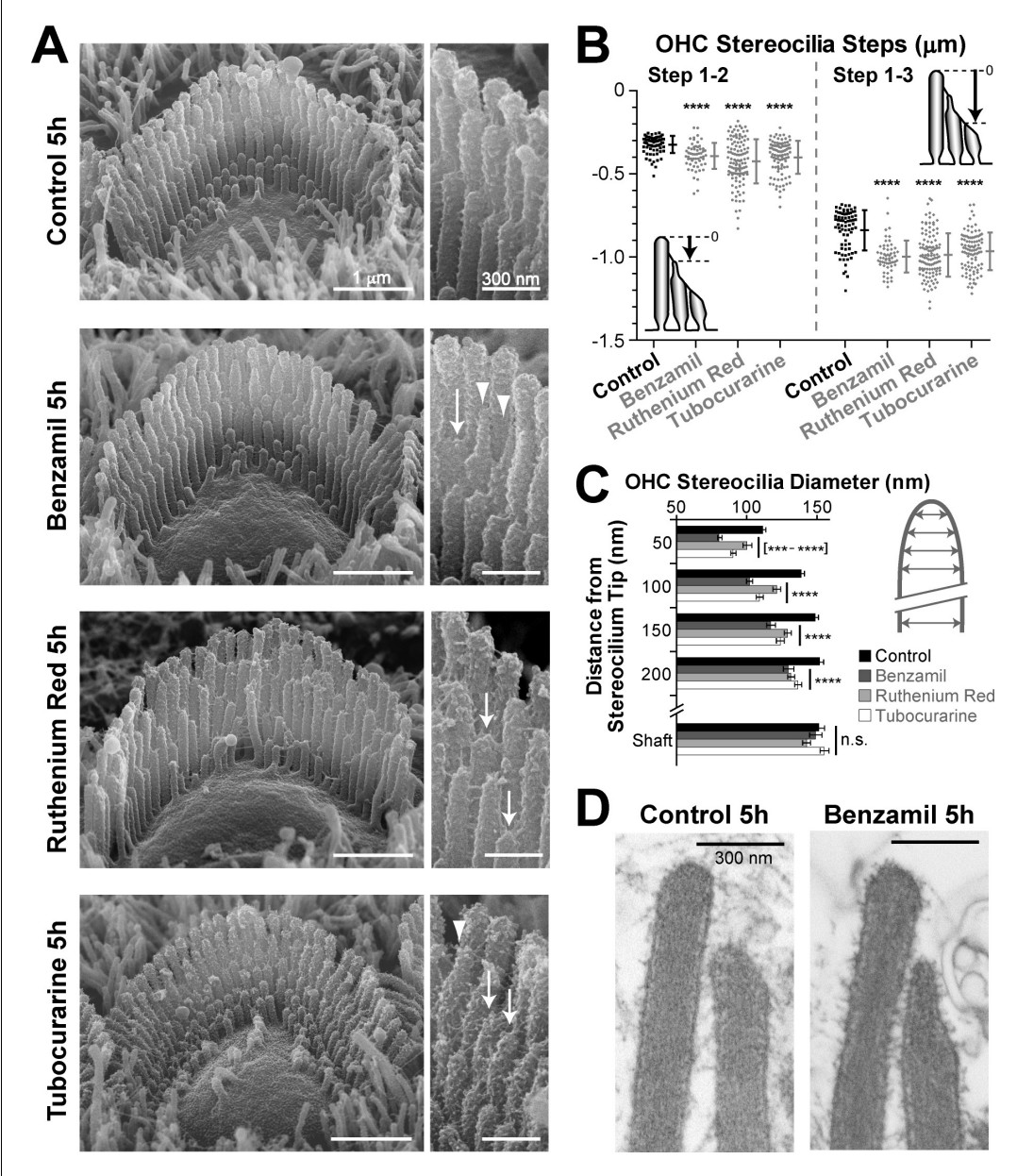

**Figure 4.** MET-dependent shortening of transducing stereocilia in OHCs initiates with thinning at the tips. (A) SEM images of mouse OHC stereocilia bundles after incubation at 37°C for 5 hr in vehicle control conditions (top) or in the presence of the MET blockers benzamil (30 μM, second row), ruthenium red (10 μM, third row) or tubocurarine (30 μM, bottom). Right panels show stereocilia at higher magnification; arrows point to retracted stereocilia and arrowheads indicate some examples of thin stereocilia tips. (B) Height differences between first- and second-row stereocilia (Steps 1–2, left) and between first- and third-row stereocilia (Step 1–3, right) in OHC bundles after 5-hr incubation in control conditions (black; n = 76) or with MET blockers (gray, n = 58–117). The inserts show the measurement procedure, which traced each stereocilium to its highest point and did not account for the shape changes at the tips. Data are from 7 to 13 cells per treatment and representative of one (ruthenium red, tubocurarine) and two (benzamil) independent series. Error bars indicate mean ± SD. Note that the staircase 'step' measurement procedure requires fewer calculations than the estimation of the absolute height of the stereocilium (as in *Figures 1D* and *2B*) and, therefore, it is more accurate for quantifying smaller changes in the staircase morphology of the bundles (see *Figure 1—figure supplement 3C,E*). (C) Diameter of second-row stereocilia at the shaft and at several positions near the stereocilia tip in OHCs after a 5-hr incubation in control conditions (black, n = 40) or with MET blockers: benzamil (dark grey, n = 44), ruthenium red (light grey, n = 42) or tubocurarine (white, n = 36). Data are from 3 to 5 cells per treatment and are shown as mean ± SE. For B and C: ***p<0.001; ****p<0.0001; n.s., non-significant (Welch's *t* tests). (D) Representative transmission electron microscopy (TEM) images of the upper part of first- and second-row stereocilia from mouse OHCs after incubation for 5 hr in control conditions (left) or in the presence of 30 μM benzamil (right). Notice the actin filaments within the abnormally thin tips of second-row stereocilia after treatment with benzamil. Age of the explants in A–D: P4-5.

*Figure 4 continued on next page*

*Figure 4 continued*

The following figure supplement is available for figure 4:

**Figure supplement 1.** MET current regulates the shape of the tips of transducing stereocilia in OHCs.

(*Figure 6D*). A similar mechanism may shape the tips of the second-row stereocilia in OHCs and IHCs in the presence of nearly saturating (but not completely blocking) concentrations of MET blockers (*Figures 4D* and *5D*). In contrast to the second-row stereocilia, tip shape changes were barely noticeable in the significantly shorter third-row OHC stereocilia after intracellular BAPTA (*Figure 6A, B*) or after MET channel blockage (*Figure 4—figure supplement 1*). However, even in these stereocilia, we observed similar shortening after both treatments (*Figures 4B* and *6E*, right panels). We concluded that, similar to the MET blockers, the increase of intracellular $Ca^{2+}$ buffering can initiate the remodeling and shortening of transducing stereocilia in auditory hair cells. Therefore, the stability of the transducing shorter row stereocilia in the auditory hair cells may be controlled by the $Ca^{2+}$ influx through the MET channels that are partially open at rest.

To study whether the remodeling of transducing stereocilia could be initiated by changes in the concentration of extracellular $Ca^{2+}$, we cultured organ of Corti explants at 37°C for 1 hr in extracellular media with different concentrations of free $Ca^{2+}$. Low extracellular $Ca^{2+}$ decreases the concentration of free $Ca^{2+}$ inside the auditory hair cell stereocilia, while high extracellular $Ca^{2+}$ increases it (*Beurg et al., 2010*), despite inhibition of MET channels in these cells by $Ca^{2+}$ ions (*Kennedy et al., 2003*). We observed shortening of transducing stereocilia in the OHCs cultured in low extracellular $Ca^{2+}$ (*Figure 6—figure supplement 1*) and thickening of the tips of second-row stereocilia in the IHCs cultured in high extracellular $Ca^{2+}$ (*Figure 6—figure supplement 1*). With these very short incubations, we were not able to detect statistically significant effects of high extracellular $Ca^{2+}$ in OHCs and low extracellular $Ca^{2+}$ in IHCs. Longer incubations were damaging to the hair cells (data not shown). We concluded that stereocilia remodeling may be initiated not only by changes in the intracellular $Ca^{2+}$ buffering but also by changes in the extracellular $Ca^{2+}$ concentration around the hair cell bundle.

## Stereocilia remodeling after disruption of the tip links

Although it has been previously reported that the loss of a tip link results in remodeling of the wedge-shaped tip of a stereocilium into a round-shaped tip (*Kachar et al., 2000*), this effect has generally been associated with the loss of tip link tension and not the loss of the resting MET current (see for example: [*Rzadzinska et al., 2004*]). In addition, all these studies focused on the tips of stereocilia, leaving potential stereocilia shortening uninvestigated. Therefore, we re-examined the effects of tip link breakage on stereocilia shape. We disrupted the tip links with extracellular $Ca^{2+}$-free medium supplemented with BAPTA, as previously described (*Indzhykulian et al., 2013*). After BAPTA treatment for 15 min, we started to observe shortening of the second and third row stereocilia in OHCs, which became obvious 1 hr later (*Figure 7A*). At several time points of recovery after BAPTA treatment, quantitative measurements revealed a significant decrease in the height of shorter row stereocilia (*Figure 7C*). Similar to the effects of MET channel blockers (*Figure 1D*), the absolute heights of the tallest OHC stereocilia **were** not affected after BAPTA and throughout recovery (*Figure 7B*), indicating the selective shortening of only transducing stereocilia. However, we did not observe abnormally thin stereocilia tips in the transducing stereocilia of OHCs after the treatment with extracellular BAPTA (*Figure 7A*, insets). This is expected because tip link disruption with BAPTA eliminates the MET current completely, in contrast to the experiments with non-saturating concentrations of MET blockers that may result in a $Ca^{2+}$ gradient across the stereocilium diameter and the preferential remodeling of peripheral actin filaments (*Figure 6G*). The regeneration of tip links led to the regrowth of transducing shorter row stereocilia by 6 hr of recovery (*Figure 7A,C*), when ~ 70% of the MET current has reappeared (*Indzhykulian et al., 2013*). Thus, there is only a relatively short time window after BAPTA treatment (less than 6 hr) when the shortening of transducing stereocilia in the OHCs could be detected. A similar but less prominent shortening of transducing stereocilia with their subsequent recovery was observed in the IHCs after treatment with extracellular BAPTA (data not shown). As expected, the MET blocker benzamil inhibited stereocilia re-growth

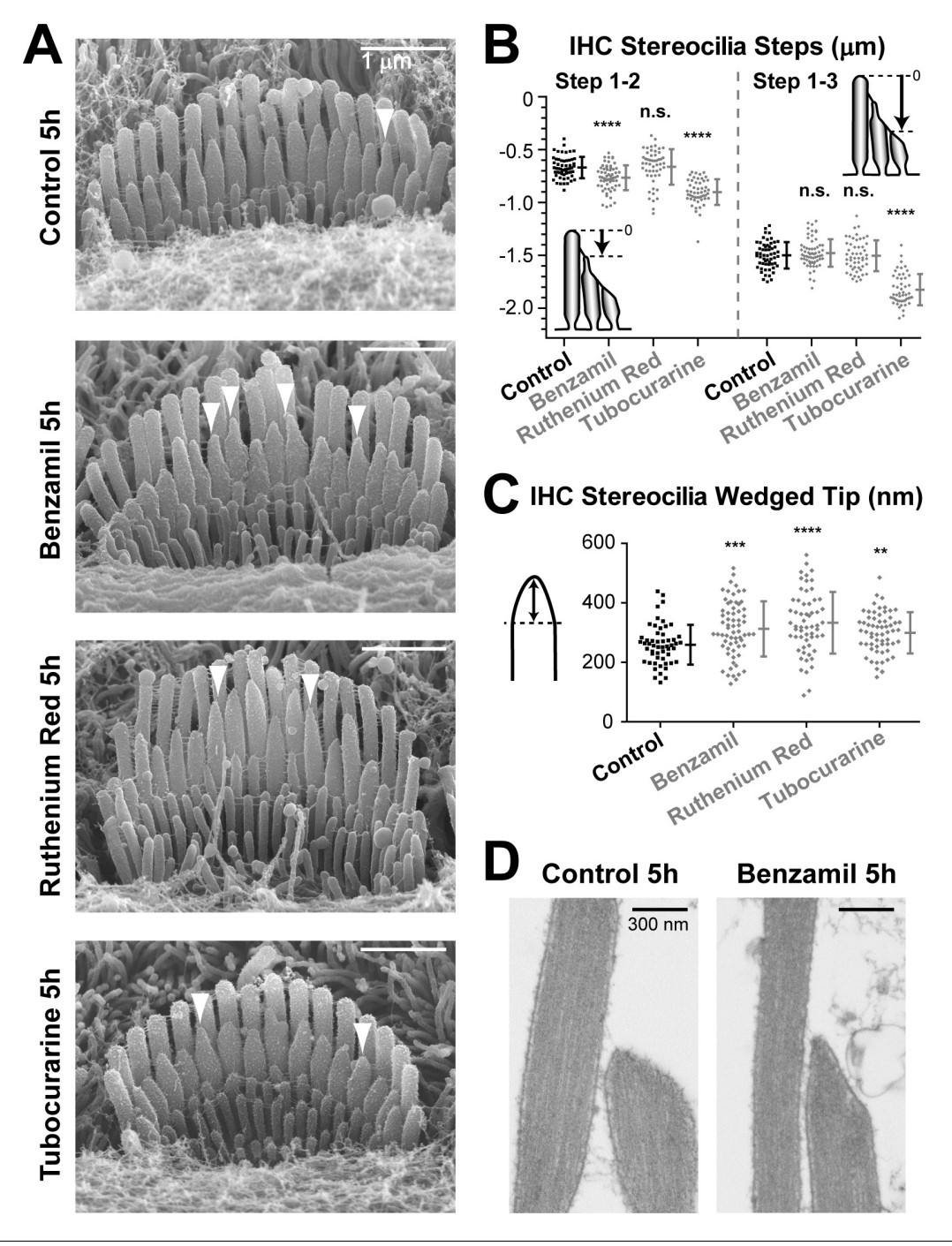

**Figure 5.** MET current regulates the shape of transducing stereocilia in IHCs. (**A, B**) Representative SEM images (**A**) and quantification of height differences between stereocilia rows (**B**) of IHC stereocilia bundles after incubation at 37°C for 5 hr in vehicle control conditions or in the presence of MET blockers: benzamil (30 µM), ruthenium red (10 µM) or tubocurarine (30 µM). Panel layouts are identical to *Figure 4A,B*. (**B**) Staircase 'step' measurements: control (n = 57), benzamil (n = 62), ruthenium red (n = 52), and tubocurarine (n = 51). Data are from 8 to 12 cells per treatment. (**C**) Heights of the wedged tips (left cartoon) from individual IHC stereocilia in the second row of the bundle after incubation for 5 hr in vehicle control conditions (black, n = 54) or in the presence of MET blockers (gray, n = 59–66). Data are from 3 to 5 cells per treatment. In **B** and **C**: Error bars indicate mean ± SD. **p<0.01; ***p<0.001; ****p<0.0001; n.s., non-significant (Welch's *t* tests). Age of explants in all panels: P4-5. (**D**) Representative TEM images of the tips of second-row stereocilia in IHCs after incubation for 5 hr in vehicle control conditions (left) or in the presence of 30 µM of benzamil (right).

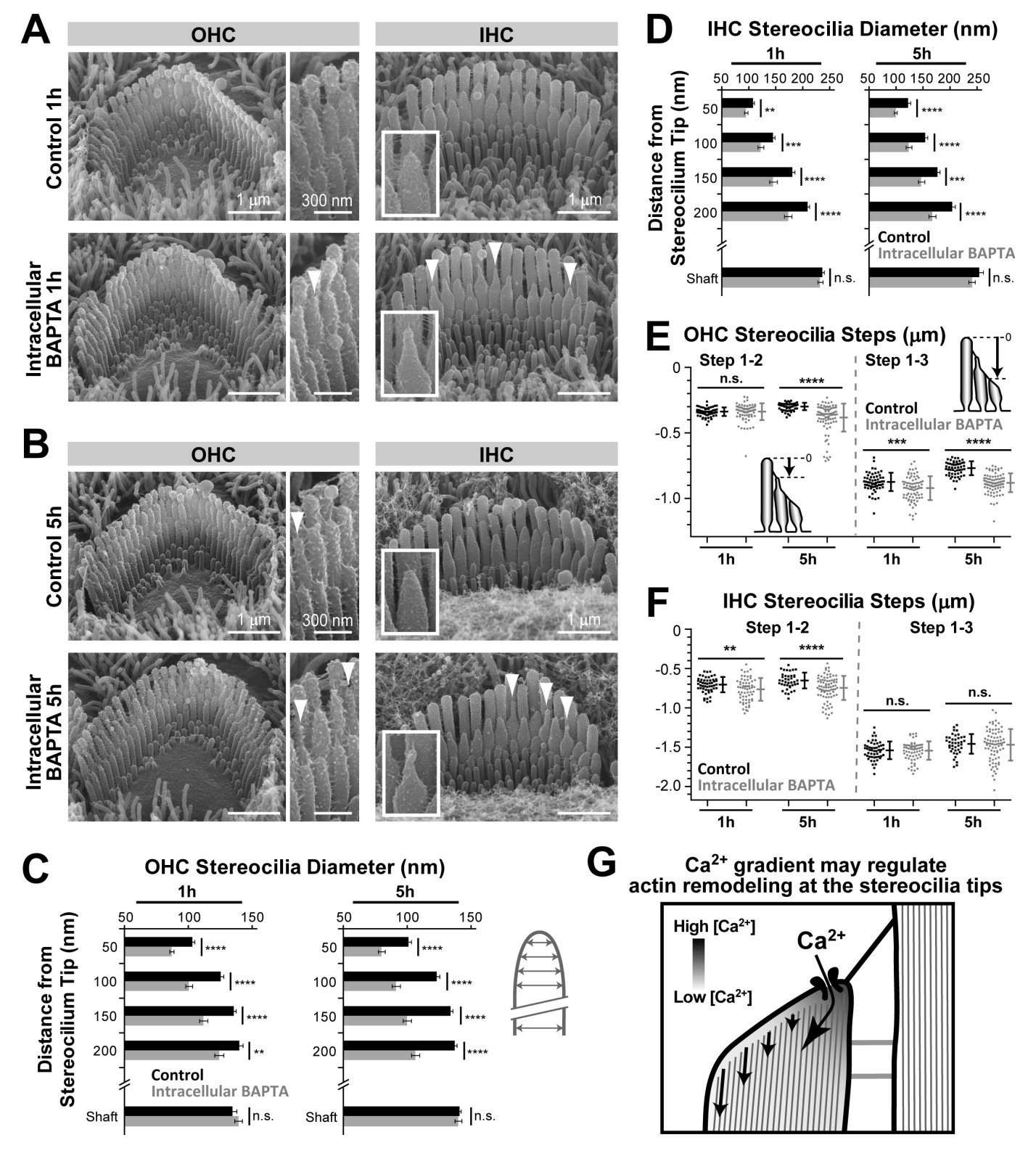

**Figure 6.** Increased intracellular Ca$^{2+}$ buffering leads to thinning and shortening of transducing stereocilia in mouse auditory hair cells. (**A and B**) SEM images of mouse OHC (left and middle) and IHC (right) stereocilia bundles after incubation for 1 hr (**A**) or 5 hr (**B**) at 37°C in vehicle-control conditions (top) or in the presence of 20 μM of the membrane-permeable BAPTA-AM (bottom). Middle panels and insets show higher magnification images of OHC and IHC stereocilia, respectively. Arrowheads indicate examples of stereocilia with abnormally thin tips. (**C and D**) Stereocilia diameter at the shaft

*Figure 6 continued on next page*

*Figure 6 continued*

and at several regions near the tip in second-row stereocilia from OHCs (**C**) and IHCs (**D**) cultured in vehicle-control conditions (black) or with BAPTA-AM (gray) for 1 hr or 5 hr (n = 41–47). Data are from 3 to 4 cells per treatment and are shown as mean ± SE. (**E and F**) Height differences between the first and second (left), and first and third (right) rows of stereocilia in OHCs (**E**) or IHCs (**F**) cultured in vehicle-control conditions (black, n = 54–60, OHCs; n = 40–52, IHCs) or with BAPTA-AM for 1 hr or 5 hr (gray; n = 78–85, OHCs; n = 64–79, IHCs). Insert cartoons in *E* clarify that we measured the length of a stereocilium to its highest point and did not account for the shape changes at the tips. Data are from 8 to 10 cells per treatment. Error bars indicate mean ± SD. For **C–F**: Data shown are from a single series of experiments, representative of three independent series. **p<0.01; ***p<0.001; ****p<0.0001; n.s., non-significant (Welch's *t* tests). Age of explants in **A–F**: P4 plus 1 or 5 hr of culturing. (**G**) Schematic diagram illustrating the intracellular $Ca^{2+}$ gradient at the tip of a transducing stereocilium. Higher $Ca^{2+}$ concentrations near the MET channel might prevent actin remodeling, while actin filaments further away from the channel might be more susceptible to depolymerization or other types of remodeling due to low $Ca^{2+}$ concentration.

The following figure supplement is available for figure 6:

**Figure supplement 1.** Remodeling of transducing stereocilia initiated by changes in extracellular $Ca^{2+}$ concentration.

after extracellular BAPTA treatment in both OHCs and IHCs (*Figure 7—figure supplement 1*), confirming that this re-growth is likely to be driven by the recovery of the MET current.

We also re-analyzed our previously published experiments (*Indzhykulian et al., 2013*) to examine the changes in the wedge-shaped tips of IHC stereocilia after tip link breakage. Immediately after treatment with BAPTA for 15 min at room temperature, the conical wedge at the tips of the second-row stereocilia was still present (*Figure 7D,E*). However, within ~10–20 min of recovery in normal extracellular medium and 37°C, the tips of these stereocilia became round (*Figure 7D,E*). The wedge-shaped tips of the second-row stereocilia reappeared with the regeneration of the tip links at 7 hr of recovery (*Figure 7D*). We concluded that the previously reported 'rounding' of the stereocilia tips after tip link breakage (*Kachar et al., 2000*; *Rzadzinska et al., 2004*) is consistent with our current results on the MET current-dependent remodeling of transducing stereocilia.

## Discussion

Our study provides the first experimental evidence for the role of the MET current in the maintenance of hair bundle structural stability in mammalian auditory hair cells. Somewhat similar mechanisms were hypothesized to explain the development of the remarkable staircase morphology of the hair cell stereocilia bundle (*Tilney et al. 1992*) or to explain particular mouse mutant phenotypes (*Alagramam et al., 2011*; *Caberlotto et al., 2011a*, *2011b*), but they have never been experimentally proved. In this study, remodeling at the tips of stereocilia and their subsequent retraction were observed (i) after treatment with various chemically unrelated blockers of the MET channels, (ii) after breakage of the tip links and associated loss of the MET current, and (iii) only in mechanotransducing shorter row stereocilia but not in the tallest row stereocilia. This evidence strongly suggests that stereocilia remodeling in our experiments is initiated by the reduction (or complete loss) of the resting MET current. Reproduction of these phenomena with the changes in intracellular $Ca^{2+}$ buffering or extracellular $Ca^{2+}$ concentration suggests that $Ca^{2+}$ influx might represent an essential component of the MET current that controls the stability of the transducing stereocilium. However, further experimentation is needed to determine the exact role of $Ca^{2+}$ in stereocilia remodeling.

An important limitation of our study is that it was performed in young postnatal hair cells with stereocilia bundles that are not entirely mature. Unfortunately, it is hard to determine whether the observed MET-dependent stereocilia remodeling is present in the adult auditory hair cells, because the adult hair cells do not survive in culture. However, several lines of evidence indicate that the auditory hair cells in the middle of the cochlea at P4-P5 may demonstrate phenomena that are present in older hair cells. First, these hair cells exhibit well-developed MET current (*Waguespack et al., 2007*; *Lelli et al., 2009*). Second, the actin core of stereocilia in these cells seems to be already stable after a period of initial growth, while actin remodeling at the stereocilia tips is qualitatively similar to that in adult cells (*Zhang et al., 2012*; *Drummond et al., 2015*; *Narayanan et al., 2015*). Third, this developmental age seems to coincide with the switch of myosin 15a isoforms that underlies the transition from predominant growth to the maintenance of the hair bundle (*Fang et al., 2015*).

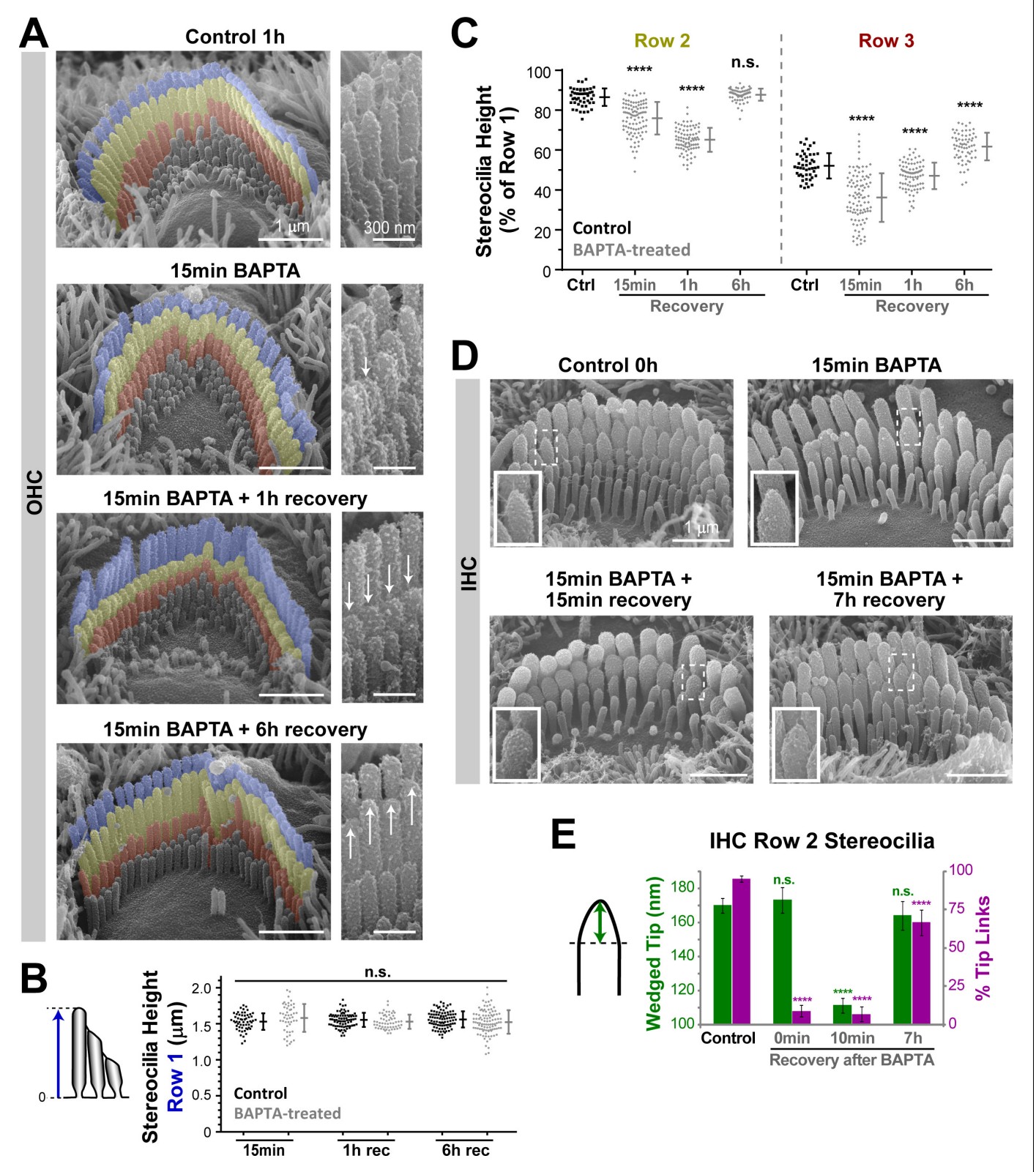

**Figure 7.** Disruption of tip links leads to remodeling of the transducing shorter row stereocilia. (**A**) Representative false-colored SEM images of OHC stereocilia bundles in mouse organ of Corti explants incubated in control conditions for 1 hr (top), immediately after treatment with BAPTA-buffered Ca$^{2+}$-free solution for 15 min (second panel), and after 1 hr (third panel) and 6 hr (bottom) recovery in Ca$^{2+}$-containing culture medium. Panels on the right show OHC stereocilia at higher magnification. The arrows point to shortened or regrown stereocilia. (**B** and **C**) Absolute heights of individual

*Figure 7 continued on next page*

*Figure 7 continued*

stereocilia in the first row (**B**), and relative heights of the second (left) and third (right) rows (**C**), in mouse OHC bundles at different times of recovery after BAPTA (gray) or in control conditions (black). Error bars indicate mean ± SD. ****p<0.0001; n.s., non-significant (Welch's *t* tests). Data are from 6 to 11 cells per time point and representative of two independent series. (**D**) SEM images of representative mouse IHC bundles before (top left), immediately after treatment with BAPTA-buffered $Ca^{2+}$-free medium for 15 min (top right), and after recovery periods of 15 min (bottom left) and 7 hr (bottom right). Insets show higher magnification views of the tips of second-row stereocilia. (**E**) The height of the wedged tips (left cartoon) of second-row stereocilia (green) and the percentage of tip links (magenta) before (Control) and at 0 min, 10 min and 7 hr of recovery after treatment with BAPTA. Quantifications of the wedged tip size and number of tip links were performed in the same IHCs. n = 33–105 stereocilia (from 4 to 12 cells) per time point. Pooled data from seven independent series, shown as mean ± SE. *p<0.05; ****p<0.0001 (Student's *t* tests). Age of explants: P4.

The following figure supplement is available for figure 7:

**Figure supplement 1.** Regrowth of stereocilia after tip link regeneration is inhibited by MET current blockage.

---

Fourth, we observed the MET-dependent remodeling of transducing stereocilia also in rat and mouse OHCs and IHCs at P6 (*Figure 1—figure supplement 1* and data not shown). Last but not least, the developmental growth of stereocilia bundles in the mid-cochlear OHCs is already at the plateau at P4 (*Kaltenbach et al., 1994*), while the growth of IHC stereocilia bundles continues until ~P18 (*Kaltenbach et al., 1994*; *Peng et al., 2009*). Therefore, the MET-dependent stereocilia remodeling observed in our study may reflect basic mechanisms controlling the height and the shape of transducing stereocilia after their initial growth.

$Ca^{2+}$-dependent mechanisms of actin core remodeling are likely to operate at the tips of stereocilia, where MET channels are in close proximity to the proteins regulating stereocilia growth (*Belyantseva et al., 2005*; *Beurg et al., 2009*). It is generally believed that most of the $Ca^{2+}$ entering through the MET channels is extruded at the stereocilium level by the plasma membrane $Ca^{2+}$ ATPase type 2 (PMCA2), which effectively 'shields' a stereocilium from the rest of the cell in both non-mammalian (*Lumpkin and Hudspeth, 1998*; *Yamoah et al., 1998*; *Dumont et al., 2001*) and mammalian (*Dumont et al., 2001*; *Beurg et al., 2010*) hair cells. The exact gradient of free $Ca^{2+}$ concentration in the stereocilium and, hence, the concentration at the very tip, also depend on mobile and fixed intracellular $Ca^{2+}$ buffers (*Lumpkin and Hudspeth, 1998*; *Beurg et al., 2010*). In this study, all experiments aimed to reduce intrastereociliar $Ca^{2+}$ resulted in shortening of the transducing stereocilia or their thinning at the tips, while all experiments designed to recover or increase intrastereociliar $Ca^{2+}$ caused re-growth of transducing stereocilia or their thickening at the tips. Observed quantitative (but not qualitative) differences in the MET-dependent remodeling of stereocilia in the IHCs and OHCs can be easily explained by available data. The expected larger resting $Ca^{2+}$ influx through the MET channels in OHCs (*Beurg et al., 2010*) should be extruded more effectively by PMCA2 that is expressed in OHC stereocilia at a higher density than in IHCs (*Chen et al., 2012*). Therefore, the blockage of the MET channels in OHCs is expected to produce a larger drop in the resting $Ca^{2+}$ concentration at the very tips of transducing stereocilia, causing a more prominent remodeling of these stereocilia in OHCs. In addition, the larger diameter of the second-row stereocilia in IHCs may promote a more prominent $Ca^{2+}$ gradient across the diameter of a stereocilium (see *Figure 6G*) and, hence, produce larger MET- and $Ca^{2+}$-dependent changes to the shape of stereocilia tips in IHCs than in OHCs. It is harder to explain why, after MET channel blockage, the third-row stereocilia in OHCs shorten but do not thin at the tips as prominently as do the second-row stereocilia (see *Figure 4—figure supplement 1*). Unfortunately, currently available techniques for the imaging of $Ca^{2+}$ transients in cochlear hair cell stereocilia (*Beurg et al., 2009*, *2010*; *Delling et al., 2016*) are limited to non-ratiometric imaging (which cannot determine the actual concentration of free $Ca^{2+}$) and are too crude to resolve $Ca^{2+}$ gradients across a stereocilium diameter. There are essential molecules that are expressed differently in the auditory hair cell bundle between the first- and second-row stereocilia (*Furness et al., 2013*; *Fang et al., 2015*; *Ebrahim et al., 2016*) but, perhaps, also between the third and second rows. The latter differences have never been quantitatively investigated in the cochlear hair cells. It is worth mentioning that, despite the remarkable stability of stereocilia, the processes underlying this stability are physiologically vulnerable. Culturing organ of Corti explants at a room temperature of 25°C results in a less prominent MET-dependent stereocilia remodeling and initiates disruption of the hair bundle

morphology (data not shown). Therefore, we avoided commonly used techniques for the manipulation of intracellular $Ca^{2+}$, such as the application of $Ca^{2+}$ ionophores or the inhibition of PMCA2, because they could be deleterious to the hair cells incubated for several hours at 37°C.

Our very limited understanding of stereocilia development and maintenance does not yet allow the proposal of a particular molecular mechanism for the MET-dependent stereocilia remodeling observed in this study. First, the $Ca^{2+}$ influx through the MET channels may have a variable effect on different actin isoforms. The stereocilia core contains both $\beta$- and $\gamma$- isoforms of actin (*Furness et al., 2005*; *Perrin et al., 2010*). Although both these isoforms are distributed along the entire length of the stereocilium, variations in the ratio between isoforms are observed (*Perrin et al., 2010*) and, in fact, exposure to noise leads to visible changes in the ratio between these two isoforms at the stereocilia tips (*Belyantseva et al. 2009*). When bound to $Ca^{2+}$, $\gamma$-actin exhibits slower polymerization and depolymerization kinetics than $\beta$-actin (*Bergeron et al., 2010*). Therefore, $Ca^{2+}$ influx through MET channels may stabilize the actin core and even shift the equilibrium toward filament growth in the areas with an increased ratio of $\gamma$- to $\beta$- actin. Second, at least some proteins controlling actin dynamics in the stereocilia are $Ca^{2+}$ sensitive. For example, the actin-bundling protein plastin 1 (a homologue of fimbrin) is expressed in hair cell stereocilia (*Flock et al., 1982*; *Taylor et al., 2015*) and contains two EF hand $Ca^{2+}$-binding sites (*Lin et al., 1994*). Plastin 1 knockout mice exhibit progressive hearing loss and stereocilia width abnormalities (*Taylor et al., 2015*). Members of the gelsolin family enhance actin dynamics upon an increase of intracellular $Ca^{2+}$ (*Kinosian et al., 1998*; *Revenu et al., 2007*) and are also present in hair cell stereocilia (*Mburu et al., 2010*; *Olt et al., 2014*). The exact function of these and other potential $Ca^{2+}$-sensitive regulators of actin in the hair cell stereocilia are yet to be determined. Third, the $Ca^{2+}$ influx may affect various myosin motors expressed in the stereocilium, similarly to the proposed effects of $Ca^{2+}$ on the myosin-based adaptation motor (*Gillespie and Cyr 2004*). Particularly interesting are myosin 15a and myosin 3. Both these myosins are involved in stereocilia length regulation by delivering their cargoes, whirlin (*Belyantseva et al., 2005*), Eps8 (*Manor et al., 2011*; *Zampini et al., 2011*), Eps8L2 (*Furness et al., 2013*), espin-1 (*Salles et al., 2009*) and ESPNL (*Ebrahim et al., 2016*) to the tips of stereocilia. Some of these proteins—the long isoform of myosin 15a (*Fang et al., 2015*), Eps8L2 (*Furness et al., 2013*) and ESPNL (*Ebrahim et al., 2016*)—are expressed predominantly at the tips of transducing stereocilia in mammalian auditory hair cells. Knockout or mutant mice with functional deficiencies in these proteins exhibit selective disassembly of stereocilia in the shortest but not tallest rows of the auditory hair cell bundles (*Furness et al., 2013*; *Fang et al., 2015*; *Ebrahim et al., 2016*). Thus, these molecules are also good candidates for the molecular machinery involved in the MET-dependent stereocilia remodeling observed in our study. Finally, the actin-severing proteins AIP1 and ADF are thought to be responsible for actin disassembly at the stereocilia tips and for balancing of the continuous incorporation of new actin monomers to the tips (*Narayanan et al., 2015*). Mice lacking ADF or expressing a mutant AIP1 exhibit defects in the hair bundle morphology (*Narayanan et al. 2015*) that are very similar to the ones observed after the blockage of MET channels in our experiments. However, it is yet unknown whether AIP1 and ADF deficiencies influence the stereocilia actin core directly or secondarily to the loss of MET current.

Independent of the molecules involved, our data demonstrate a functional link between the resting MET current and stereocilia remodeling. After the initial report on the incorporation of exogenous $\beta$-actin into the stereocilia of young postnatal rats (*Schneider et al., 2002*), it was hypothesized that the stereocilia actin core is maintained through the continuous treadmill of actin (*Rzadzinska et al., 2004*). A similar relatively fast actin remodeling was demonstrated in zebrafish stereocilia but without evidence of treadmilling (*Hwang et al., 2015*). On the other hand, several independent groups have now established that in adult and young mammalian and non-mammalian hair cells, the active actin remodeling occurs only in a small (~0.5 µm) region at the tips of stereocilia but not along their shafts (*Zhang et al., 2012*; *Drummond et al., 2015*; *Narayanan et al., 2015*). Our data reconcile these different points of view on the stability of stereocilia actin core, at least in the transducing stereocilia of mammalian hair cells. Apparently, in the presence of a normal resting current through the MET channels, actin remodeling is limited to the tips of stereocilia. However, when the $Ca^{2+}$ concentration inside the stereocilia changes significantly after blocking or unblocking the MET channels, the equilibrium is shifted and stereocilia start to retract or re-grow respectively, expanding the area of active actin remodeling. Interestingly, the areas of incorporation of fluorescently labelled actin to the tips of stereocilia in the second row of IHCs are larger than that in the

tallest row stereocilia and vary significantly between individual stereocilia (*Narayanan et al., 2015*), similarly to the variations of MET-dependent $Ca^{2+}$ influx into these stereocilia (*Beurg et al., 2009*). Furthermore, deletions in the genes encoding currently known components of the MET machinery— TMC1/TMC2 (see Supplemental Figure 4B in [*Kawashima et al., 2011*]), TMHS (see Fig. 1D in [*Xiong et al., 2012*]), TMIE (see Fig. 5A-B in [*Zhao et al., 2014*])—result in the loss of MET current and changes of the hair bundle morphology that seem to be limited to the shortest but not tallest row stereocilia in the auditory hair cells. These abnormalities are very similar to the changes that we observed in this study after blocking MET channels. Thus, we believe that the changes of stereocilia bundle morphology in these mouse mutants are likely initiated by the loss of the MET current.

What is the physiological significance of the MET-dependent stereocilia remodeling? It is unlikely to drive the initial formation of the hair bundle due to the relatively low amplitude of the MET current in the first postnatal days (*Waguespack et al., 2007*; *Lelli et al., 2009*) and the rather normal stereocilia bundle formation in mutant mice lacking MET currents (*Kawashima et al., 2011*; *Xiong et al., 2012*; *Zhao et al., 2014*). However, this mechanism may be essential for maintenance and/or fine tuning of the staircase shape of the mature hair bundle after substantial MET current has developed. Additionally, our data may represent an exaggerated manifestation of stereocilia remodeling at the stereocilia tips that operates at a faster time scale as compared to the overall changes of stereocilia height. If this is the case, then the MET-dependent remodeling of stereocilia tips may also contribute to processes such as the dynamic regulation of tip link tension.

The dynamic control of stereocilia remodeling by the $Ca^{2+}$ influx through the MET channels works in parallel with any other mechanisms that are responsible for tensioning the tip links, such as the operation of myosin-based molecular motors (reviewed in [*Gillespie and Cyr, 2004*]). Our data indicate that, even after the 24 hr incubation with MET channel blockers and the significant shortening of the transducing stereocilia, the resting tension in the transduction machinery is still present, which allows FM1-43 accumulation inside the OHCs immediately after blocker washout (*Figure 1*). Furthermore, this tension is likely to be essential for the recovery of the MET current and the stereocilia regrowth after washing out the MET blockers (*Figure 3*). In fact, it is tempting to speculate that the upward force of myosin motors may eventually determine the exact staircase architecture of a stereocilia bundle, which represents one of the most enigmatic problems in hair cell biology. The height of a transducing stereocilium may be set by a delicate equilibrium between the modulatory influence of the $Ca^{2+}$ influx on actin assembly/disassembly and the rate of delivery of essential molecules to the tip, which is likely to be inversely proportional to the height of a stereocilium. By setting a certain tension of the tip link and MET current at rest, the myosin motors may set this equilibrium at a precise stereocilium height and determine, for example, the final heights of the second- and third-row stereocilia when they re-grow after the washout of MET blockers (*Figure 3*). Independent of whether these speculations are true or not, the MET-dependent stereocilia remodeling demonstrated in our study is likely to represent an important mechanism for maintenance and repair of the hair bundles in non-regenerating mammalian auditory hair cells.

## Materials and methods

### Organ of Corti explants

Organ of Corti explants were isolated from C57BL/6 (RRID:IMSR_JAX:000664, Jackson Laboratories, Bar Harbor, ME) or CD1 (Charles River Laboratories, Wilmington, MA) wild-type mice (both male and female) at postnatal days 4 (P4) through P6, or from Sprague-Dawley rats (RRID:RGD_734476, Charles River Laboratories) at P6. The explants were held by two flexible glass fibers (~1–2 cm in length) glued to the bottom of plastic Petri dishes (Electron Microscopy Sciences) using the silicone elastomer Sylgard (World Precision Instruments). Explants were cultured at 37°C and 5% $CO_2$ in DMEM (Invitrogen, Carlsbad, CA) either alone or supplemented with 7% fetal bovine serum (FBS, Atlanta Biologicals, Flowery Branch, GA) and 10 μg/mL ampicillin (Calbiochem, San Diego, CA). Each experiment was typically performed for at least two to three independent series. Each series consisted of comparisons between tissue samples (organs of Corti) from littermates, including comparisons between the two ears of the same animal (i.e. control vs. treated). All animal procedures were approved by the Institutional Animal Care and Use Committee (IACUC) at the University of Kentucky (protocol 00903M2005).

## FM1-43 uptake

Freshly isolated organ of Corti explants were incubated for 30 s in ice-cold $Ca^{2+}$-containing standard Hank's balanced salt solution (HBSS, catalog number 14025, Invitrogen) supplemented with 6 μM of FM1-43FX in the absence or presence of various blockers of MET channels (see below). Then, the explants were rinsed thoroughly with cold $Ca^{2+}$-containing HBSS and fixed in 4% paraformaldehyde (PFA) solution for 30 min. Immediately after fixation, the samples were rinsed with HBSS and imaged. For the recovery experiments (*Figure 1B*), the organ of Corti explants were cultured for 24 hr in the presence of benzamil (30 μM) and exposed to FM1-43 for 30 s either before or after drug washout. Samples were rinsed thoroughly with HBSS and imaged immediately after. Imaging was performed using an upright Olympus BX51WI microscope equipped with a 40X (0.8 NA) LUMPlanFL water-immersion objective and spinning disc confocal attachment (DSU). In all experiments, the osmolarity of HBSS was adjusted to 310 mOsm with ~20 mM of D-glucose.

## MET channel blocking

Freshly isolated organ of Corti explants were incubated (cultured) in FBS/ampicillin-supplemented DMEM at 37°C and 5% $CO_2$ for 5–32 hr with or without the following MET channel blockers: benzamil (5.5, 10 or 30 μM), amiloride (100 μM), ruthenium red (10 μM) and tubocurarine (30 μM) (Sigma-Aldrich, St. Louis, MO). The maximum concentrations used were chosen based on previously reported dose-response curves in order to block 75–90% of the MET current (*Rüsch et al., 1994*; *Farris et al., 2004*) and to minimize potential deleterious effects of these drugs during long-term incubations. The control explants were incubated in parallel in the same medium but without drugs, except for the vehicle control for benzamil that included 0.05% of DMSO (Molecular Probes, Eugene, OR). The samples treated for 5 hr were briefly rinsed with HBSS and fixed immediately after incubation. After long incubations for 24–32 hr, the samples were first placed in HBSS (with or without the same MET blocker that was used in the experiment) and observed with an upright microscope (E600FN, Nikon). The fibrous material from tectorial membrane outgrowth was gently removed with a ~2–4 μm suction pipette mounted on a micromanipulator (MHW-3, Narishige, Tokyo, Japan). Then, the explants were fixed for electron microscopy.

## Disruption of tip links

Freshly isolated mouse organ of Corti explants were first rinsed with standard HBSS. Next, the explants were incubated for 15 min at room temperature in $Ca^{2+}$-free HBSS (catalog number 14175, Invitrogen) supplemented with 5 mM of $Ca^{2+}$ chelator, 1,2-bis(o-aminophenoxy)ethane-N,N,N',N'-tetraacetic acid, BAPTA (Sigma-Aldrich) and 0.5 mM or 0.9 mM of $Mg^{2+}$. After incubation, the explants were rinsed with the standard $Ca^{2+}$-containing HBSS and allowed to recover in FBS/ampicillin-supplemented DMEM at 37°C and 5% $CO_2$ for different periods of time up to 6 hr. At the end of the recovery period or immediately after incubation with BAPTA, the explants were fixed for electron microscopy (see below).

## Alterations of intracellular $Ca^{2+}$ buffering

The membrane-permeable $Ca^{2+}$ chelator BAPTA-AM (Molecular Probes) was pre-mixed with a 20% Pluronic F-127 solution in DMSO (Molecular Probes). Freshly isolated P4 mouse organ of Corti explants were incubated in FBS/ampicillin-supplemented DMEM at 37°C and 5% $CO_2$ for 1 to 5 hr in the presence of 20 μM BAPTA-AM (in Pluronic/DMSO) or in vehicle control conditions (0.1% of the Pluronic/DMSO solution). At the end of the incubation, the explants were rinsed with standard HBSS and placed in cold fixative.

## Alterations of extracellular $Ca^{2+}$ concentrations

Freshly isolated P4 mouse organ of Corti explants were fixed after 1 hr incubation at 37°C and 5% $CO_2$. Longer incubations, especially with high extracellular $Ca^{2+}$, were found to be deleterious to the hair cells. Some explants were incubated in DMEM alone (~1.8 mM $Ca^{2+}$) or in DMEM supplemented with 1.7 mM BAPTA to lower the free $Ca^{2+}$ concentration (~100 μM $Ca^{2+}$). Other explants were incubated in FBS/ampicillin-supplemented DMEM, alone (~1.85 mM $Ca^{2+}$) or supplemented with 5 mM $CaCl_2$ (~6.85 mM $Ca^{2+}$) or 10 mM $CaCl_2$ (~11.85 mM $Ca^{2+}$).

## Scanning electron microscopy

Organ of Corti explants were fixed for at least 2 hr in a mixture of 3% PFA and 3% glutaraldehyde in 0.1 M sodium cacodylate buffer, pH 7.4 (Electron Microscopy Sciences, Hatfield, PA) supplemented with 2 mM of $CaCl_2$ (Sigma-Aldrich). The samples were rinsed with distilled water, dehydrated through a graded series of ethanol, critical point dried from liquid $CO_2$ (EMS 850, Electron Microscopy Sciences), sputter-coated with 5-nm platinum (Q150T, Quorum Technologies, Guelph, Canada), and imaged with a field-emission scanning electron microscope (Helios Nanolab 660, FEI, Hillsboro, OR). To avoid damage of the sample with the electron beam, imaging was performed at a small working distance (~4 mm), which improved signal-to-noise ratio and allowed the use of smaller apertures. To accurately quantify the height of individual stereocilia in the different rows of a hair bundle, we obtained images of the same bundle from different angles, including views from the lateral ('back') and medial ('front') sides of the bundle. *Figure 1—figure supplement 3* describes the methods of quantification. Samples with any signs of failed SEM preparation were discarded. These signs include: (i) fused or curved stereocilia due to encountering surface tension; (ii) lack of tip links in the untreated control samples; and (iii) mounting errors that would not allow imaging of the same bundle from the 'front' and 'back' sides. Measurements were performed by an examiner blind to the experimental conditions using ImageJ (RRID:SCR_003070).

## Tip link count

A 'tip' link was defined as a link that extends obliquely from the top of a lower row stereocilium to the side of a taller stereocilium in the direction of mechanosensitivity of the bundle. Any other link originating at the hemisphere of the tip of a shorter stereocilium was not considered as a tip link. See (*Indzhykulian et al., 2013*) for more details.

## Transmission electron microscopy

Organ of Corti explants were fixed overnight in 3% glutaraldehyde in 0.15 M sodium cacodylate buffer (pH 7.4) at 4°C, rinsed thoroughly with sodium cacodylate buffer, and post-fixed for 1 hr in 1% tannic acid. Then, the samples were rinsed with distilled water and cryoprotected by overnight incubations in 5, 10 and 30% glycerol solutions. The explants were then placed on 3 mm copper grids and plunge frozen in liquid Freon before being transferred to 1% uranyl acetate in methanol at −90°C in a Leica EM AFS2 freeze substitution machine. Methanol was exchanged for Lowicryl HM-20 resin and polymerized by long-wave UV radiation. All reagents were obtained from Electron Microscopy Sciences. The resin blocks were trimmed on an ultramicrotome (UC6, Leica, Wetzlar, Germany) and then milled with a focused ion beam and imaged in 'Slice and View' mode with a backscattered electron detector using the FEI Helios 660 Nanolab system.

## Acknowledgements

We thank Dr. James R Bartles and Dr. Lili Zheng for valuable comments and productive discussions, as well as for their efforts to reproduce the observed phenomena in an *in vitro* system. This study was supported by NIDCD/NIH (R01 DC014658 and R01 DC008861 to GIF) and American Hearing Research Foundation (to ACV).

## Additional information

### Funding

| Funder | Grant reference number | Author |
|---|---|---|
| National Institute on Deafness and Other Communication Disorders | R01 DC014658 | Gregory I Frolenkov |
| National Institute on Deafness and Other Communication Disorders | R01 DC008861 | Gregory I Frolenkov |
| American Hearing Research Foundation | | A Catalina Vélez-Ortega |

The funders had no role in study design, data collection and interpretation, or the decision to submit the work for publication.

### Author contributions
ACV-O, Conceived the study, Designed and performed experiments, Performed SEM imaging, Analyzed and interpreted the data, Acquired funding, Wrote the first draft of the paper, Revised all versions of the manuscript; MJF, Performed blind measurements of stereocilia dimensions from SEM images, Assisted with preparing the manuscript; AAI, Performed the first series of experiments on the changes of stereocilia tip shape after breakage of the tip links, Assisted with preparing the manuscript; JMG, Prepared all samples for FIB-SEM using fast-freezing, freeze-substitution, and low temperature embedding, Assisted with preparing the manuscript; GIF, Conceived the study, Supervised the project, Acquired funding, Designed experiments, Performed SEM and FIB-SEM imaging, Analyzed and interpreted the data, Reviewed and edited all versions of the manuscript

### Author ORCIDs
A Catalina Vélez-Ortega, http://orcid.org/0000-0001-9157-8390
Gregory I Frolenkov, http://orcid.org/0000-0002-9810-5024

### Ethics
Animal experimentation: This study was performed in strict accordance with the recommendations in the Guide for the Care and Use of Laboratory Animals of the National Institutes of Health. All animal procedures were approved by the Institutional Animal Care and Use Committee (IACUC) at the University of Kentucky (protocol 00903M2005).

# Additional files

### Major datasets
The following dataset was generated:

| Author(s) | Year | Dataset title | Dataset URL | Database, license, and accessibility information |
|---|---|---|---|---|
| Vélez-Ortega AC, Freeman MJ, Indzhykulian AA, Grossheim JM, Frolenkov GI | 2017 | Data from: Mechanotransduction current is essential for stability of the transducing stereocilia in mammalian auditory hair cells | http://dx.doi.org/10.5061/dryad.5cp90 | Available at Dryad Digital Repository under a CC0 Public Domain Dedication |

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
