## [Decision Letter]

Thank you for submitting your article "Mechanotransduction current is essential for stability of the transducing stereocilia in mammalian auditory hair cells" for consideration by *eLife*. Your article has been reviewed by three peer reviewers, and the evaluation has been overseen by a Reviewing Editor (Jeremy Nathans) and Richard Aldrich as the Senior Editor.

The reviewers have discussed the reviews with one another and the Reviewing Editor has drafted this decision to help you prepare a revised submission. Three experts reviewed your manuscript, and their assessments, together with my own, form the basis of this letter. I am including the three reviews at the end of this letter, as there are specific and useful suggestions in them. As you will see, all of the reviewers were impressed with the importance and novelty of your work.

*Reviewer #1:*

This is a compelling contribution from the Frolenkov laboratory. It has been speculated for some time that mechanotransduction in hair cells might be critical to regulate hair bundle morphogenesis. However, clear experimental evidence for this model has been lacking. As outlined in the Discussion of this manuscript, there has also been substantial controversy regarding the stability of the actin filaments in hair cells with some reports providing evidence for treadmilling of actin within stereocilia, while others provided evidence that actin dynamics might be more restricted to tips of stereocilia. It has been speculated that treadmilling is perhaps obvious during development but not in mature stereocilia.

The manuscript by Velez-Ortega provides insights on the effect of transduction on stereocilia and on actin stability. Recent studies of hair cells in mice lacking functional transduction channels already suggested that the typical staircase pattern of stereocilia can be established during development even without functional transduction channels. However, closer inspection of the images in the published manuscripts suggests that stereocilia are not absolutely normal and some indication of shrinkage at tips could be observed (this was not emphasized in the manuscript and it was not their focus). The current study now demonstrates that pharmacological block of transducer channels or breakage of tip links leads to shortening of the stereocilia at their tips. Significantly, the shortening is only observed in the second and third row of stereocilia that contain mechanotransduction channels but not in the first row that does not contain transduction channels. Some data are provided that are consistent with the view that calcium might play a role in remodeling stereocilia following block of transduction.

I would like to raise several points for consideration:

1) The data are nicely presented and quantified but I think the role of calcium is somewhat speculative and overemphasized. The abstract states that constitutive calcium influx is essential for stereocilia stability. The paragraph on calcium regulation in the Results section is full of speculations. I think it is prudent to have statement like: "the data suggest that calcium influx through the transduction channel might be important" but the data on this point are soft. It would take substantially more experimentation (most of which seems currently technically not feasible) to reach the conclusion that calcium flux through the channel drive remodeling of stereocilia.

2)The authors write in several places that it is very likely that calcium influx affects the actin core of stereocilia. However, I think to demonstrate this would take further experimentation. Clearly, actin is remodeled but we do not know what the primary target of regulation is or even if calcium is the major drive. Perturbations of calcium lead to milder effects compared to perturbations of transduction. Of course, it is hard to imagine how changes in stereociliary length could occur without changes in actin. However, we do not know what the mechanistic link is. The Discussion nicely highlights several possibilities but short of experimental data this statement related to actin are prudent for the Discussion but not for other parts of the manuscript.

3) It would be nice if some experiments could be included that investigate actin more directly. Rearrangements in actin could be observed in hair cells expressing actin-GFP. Is there movement of actin throughout the bundle or just at the tips when transducer channels are pharmacologically blocked? Or do the authors have a good argument that these experiments are technically not feasible at this point?

4) The authors mention that similar phenomena have been observed in mice with mutations in TMC/LHFP5/TMIE. An analysis of one of the mutants would be a great addition to the manuscript supporting pharmacological evidence with genetic evidence.

5) The explants were cultured for 24 hours in the presence of blockers. Do the stereocilia reach at this time point a new length equilibrium or do they shrink further upon prolonged culturing? Is it difficult to extend the culture period in the presence of the blockers (cell survival?).

6) One limitation of the study is that experiments were carried out at early postnatal ages. Thus, it is not clear whether similar mechanisms maintain hair bundles in adult mice or only during development. This is a limitation of the study. The authors argue around it but I think this needs to be stated clearly without trying to argue that these are mature stereocilia.

7) One wonders about the physiological relevance of the phenomenon. Are we really looking at a mechanism that is in vivo relevant for maintaining stereocilia length on the long time scale of the experiments? An alternative view could be that the experiments reveal a different underlying physiological process that operates at a much faster time scale. Perhaps increased influx of calcium during active transduction following noise stimulation leads to local changes in actin or its interaction with the membrane that are critical to re-establish tension in the transduction machinery following channel opening thus contribution to adaptation. What is observed here is just an exaggerated manifestation of a different underlying physiological process. May be this could be discussed.

*Reviewer #2:*

This manuscript from Vélez-Ortega et al. tackles a very topical and interesting issue, for which we still know very little. Cochlear sensory hair cells rely on mechanoelectrical transduction to convey sound information to the auditory afferent fibers. One crucial characteristic of the mechanoelectrical transducer (MT) channel is that it is partially open at rest, such that a depolarizing current flows into the hair cells even in the absence of stimulation. The MT channel is a non-selective cation channel with relatively high permeability, but low conductance, for calcium. Calcium influx through the MT channel is believed to control the extent and rate of channel adaptation. In this study, the authors have proposed that the sustained influx of calcium through the MT channel is also crucial for the stability of the stereociliary bundle. Experimental manipulations that reduced the MT current or affected its open probability, resulted in the shortening and changes in shape of the shorter rows of stereocilia, which are those believed to have the MT channel. The conclusions of the paper are well substantiated by the results. All experiments are well executed in line with the excellent reputation of this lab. I have a few points for the attention of the authors:

As mentioned above, the data strongly support the conclusion. However, I am struggling to link these in vitro findings with the hypothetical in vivo scenario. The staircase-like architecture of the stereociliary bundle is mainly acquired during late embryonic and early postnatal stages of development (up to about P4). However, during this critical period the MT current and the MT channel open probability are absent or very small (onset is about P2-P3 in apical OHCs: i.e. Waguespack et al. 2007; Lelli et al. 2009). Therefore, in the in vivo situation it is unlikely that any calcium will flow into the stereocilia during the time when their precise length and width is established. This seems to suggest that calcium is not required for the formation of the stereocilia. Moreover, recent evidence from the Holt group has shown that in the Tmc1/Tmc2 double KO, in which the MT current is completely absent, the stereocilia seem normal, at least in the apical coil. I wonder whether the authors have any thoughts on how their data can be reconciled with the hypothetical in vivo scenario.

Subsection “Blockage of the MET channels leads to length dysregulation and overall shortening of transducing stereocilia”, second paragraph. Some clarification is required about the growth of stereocilia. The authors have indicated that the taller stereocilia reach their final length at about P4. This is in contrast with previous publications suggesting that final length is reached after P12 in both IHCs and OHCs (e.g. Kaltenbach et al. 1994; Peng et al. 2009).

Discussion, third paragraph. The information that the observed remodelling is affected by temperature is interesting and it seems to me to be worth showing.

*Reviewer #3:*

This is a fascinating and exciting paper that provides firm evidence that functional mechano-electrical transducer (MET) channels are required to maintain the height of the 2nd and 3rd rows of the actin-filled stereocilia that comprise the hair bundles of cochlear hair bundles. The work has been performed to an exemplary standard and is, for the most part, very convincing. Whether or not this is 'the first experimental evidence for the role of the MET current in the maintenance of hair bundle structural stability in mammalian hair cells' is somewhat debatable, unless one decides that all data obtained from transgenic mice harbouring temporally conditional alleles do not count as experimental evidence. What this paper does show though, and this is novel, is that the shortening of the rows of the transduction competent stereocilia induced by loss of MET channel activity is rapid (and therefore more likely to be a direct consequence of channel block), entirely reversible, and dependent on calcium influx.

There are a few points that the authors may want to consider:

Abstract and Introduction, second paragraph: Whilst Ca^2+^ influx through the MET channels open at rest may be considered as 'constitutive' there are also active ATPases that pump the Ca^2+^ out so is this influx really the same as Ca^2+^ loading?

Introduction, first paragraph: The hair cells in the avian basilar papilla, for example, do not regenerate or turnover unless the hearing organ is damaged and therefore have to maintain their precisely arranged stereocilia for life.

Subsection “Blockage of the MET channels leads to length dysregulation and overall shortening of transducing stereocilia”, first paragraph: Do MET channels remain functional after long-term (24h) blockage with 100 μm amiloride?

Subsection “Blockage of the MET channels leads to length dysregulation and overall shortening of transducing stereocilia”, third paragraph: Is there an explanation for why the largest effects were observed in the 3rd row of OHCs?

Figure 1: Labelling of the y-axis: It is not formally shown that these are transducing stereocilia so it may be more correct to refer to these (as in the text) as 2nd and 3rd row stereocilia.

Figure 2: Is the control IHC bundle representative? This bundle has up to 5 rows of stereocilia, the amiloride treated has 4, whilst the benzamil treated has mostly 3. Are the very short rows disappearing too and are these therefore also transduction competent?

Figure 4: It may be useful to define, as shown later in Figure 6, exactly how the height reduction is measured. Presumably the stereocilia with pointed tips indicated by the arrowheads in the benzamil and tubocurarine treated cells are not shortened? Also, the tubocurarine treated hair bundle looks exceptionally hairy. Was this observed in all samples that had been exposed to dTC?

Figure 5: The actin filaments and plasma membranes are poorly resolved. Was this image obtained from the milled samples? And were the images in Figure 4 obtained using conventional TEM? Please specify.

Subsection “Ca^2+^ influx through the MET channels controls the remodeling of transducing stereocilia”, first paragraph: To save the reader having to read the Peng et al. Neuron paper please state whether BAPTA-AM affects the open probability of the MET channel at rest.

Figure 7 and elsewhere: How is a tip link defined? There are often multiple links present at the tips of the stereocilia shown. Are these all counted as tip links irrespective of the direction they are pointing? The 7h post BAPTA treated bundle shown in the lower right hand panel of part D has remarkably few tip links and does not look representative of the data shown in the bar graph below.

Figure 4—figure supplement 1: This is a very pleasing and informative figure.

Subsection “Ca^2+^ influx through the MET channels controls the remodeling of transducing stereocilia”, last paragraph: Are long term (up to 1h) changes in the extracellular calcium around cochlear OHC hair bundles likely to occur in vivo and if so under what conditions? This may be something that could be discussed or mentioned briefly at this point.

Subsection “Stereocilia remodeling after disruption of the tip links”, first paragraph: Is there an explanation for this? Are the MET blockers simply more efficient than extracellular BAPTA? Or does this mean that stereocilia can shorten without first getting thinner at their tips?

Subsection “Stereocilia remodeling after disruption of the tip links”, last paragraph: Would these re-analysed data also be consistent with some element of tension being provided by the tips links climbing up the stereocilium?

Figure 4 onward use a measurement of stereocilia steps as opposed to heights. Is there a reason for this?

The Discussion is rather long and very speculative. Many possible mechanisms and molecules are suggested to be involved none of which have been tested. Surprisingly there is no discussion of what determines the final height of the 2nd and 3rd rows when they recover from transient MET channel block.

Discussion, fifth paragraph: Only one of the three papers cited, the Xiong et al. paper describing the Tmhs-/- mouse, appears to provide evidence for changes in the length in the transducing but not the tallest row of stereocilia. The stereocilia in the third row appears mainly to shorten, but the range of stereocilia steps between the first and second rows and second and third rows becomes extremely variable, suggesting some in the 1st and 2nd rows are shrinking and/or others may even be getting longer (see Figure S1 in Xiong et al., 2012; but note that the step size is unlikely to be in mm as stated).

Subsection “Scanning Electron Microscopy (SEM)”: graded not grated;delete 'of' from '5 nm of platinum'.

---

## [Author Response]

*Reviewer #1:*

*[…] I would like to raise several points for consideration:*

*1) The data are nicely presented and quantified but I think the role of calcium is somewhat speculative and overemphasized. The abstract states that constitutive calcium influx is essential for stereocilia stability. The paragraph on calcium regulation in the Results section is full of speculations. I think it is prudent to have statement like: "the data suggest that calcium influx through the transduction channel might be important" but the data on this point are soft. It would take substantially more experimentation (most of which seems currently technically not feasible) to reach the conclusion that calcium flux through the channel drive remodeling of stereocilia.*

Following the reviewer’s comment, we have softened our conclusions stating that “Ca^2+^ influx might represent an essential component of the MET current that controls the stability of the transducing stereocilium. However, further experimentation is needed to determine the exact role of Ca^2+^ in stereocilia remodeling.” We re- phrased also relevant descriptions of the Results to eliminate a few unnecessary speculative statements and to soften our intermediate conclusions on the role of Ca^2+^.

*2)The authors write in several places that it is very likely that calcium influx affects the actin core of stereocilia. However, I think to demonstrate this would take further experimentation. Clearly, actin is remodeled but we do not know what the primary target of regulation is or even if calcium is the major drive. Perturbations of calcium lead to milder effects compared to perturbations of transduction. Of course, it is hard to imagine how changes in stereociliary length could occur without changes in actin. However, we do not know what the mechanistic link is. The Discussion nicely highlights several possibilities but short of experimental data this statement related to actin are prudent for the Discussion but not for other parts of the manuscript.*

As suggested by the reviewer, we have removed mentioning of actin remodeling from the Results section, except in the description of the experiments that explore stereocilia actin directly, i.e. TEM imaging. We do agree with the reviewer that it is hard to imagine how changes in stereociliary length could occur without changes in actin and, therefore, we keep this idea in the manuscript.

*3) It would be nice if some experiments could be included that investigate actin more directly. Rearrangements in actin could be observed in hair cells expressing actin-GFP. Is there movement of actin throughout the bundle or just at the tips when transducer channels are pharmacologically blocked? Or do the authors have a good argument that these experiments are technically not feasible at this point?*

We do consider using transgenic mice expressing β-actin-GFP as well as using other mouse mutant models lacking different stereocilia proteins that might be involved in the MET-dependent stereocilia remodeling (see our Discussion). It is worth mentioning though one technical issue. The individual 2^nd^ and 3^rd^row stereocilia in the outer hair cells are hard to resolve by fluorescent microscopy. Meanwhile, those are the stereocilia where we see the most prominent MET-dependent remodeling.

*4) The authors mention that similar phenomena have been observed in mice with mutations in TMC/LHFP5/TMIE. An analysis of one of the mutants would be a great addition to the manuscript supporting pharmacological evidence with genetic evidence.*

We do agree that it would be nice to confirm our results in a genetic model that has disrupted mechanotransduction without any other effects on the structural proteins in the stereocilium. Unfortunately, the absence of additional effects on stereocilia proteins cannot be guaranteed for any currently available genetic model, including TMC/LHFP5/TMIE knockouts. Therefore, we believe that including these model(s) may actually decrease the convincing power of our study. The only thing that we could say is that the stereocilia phenotype in these mutant models is consistent with stereocilia remodeling after MET blockage. Therefore, we included more details referencing the exact figures in the published manuscripts that illustrate the changes of the hair bundle morphology in mice lacking TMC, LHFP5 and TMIE (see Discussion, fifth paragraph).

*5) The explants were cultured for 24 hours in the presence of blockers. Do the stereocilia reach at this time point a new length equilibrium or do they shrink further upon prolonged culturing? Is it difficult to extend the culture period in the presence of the blockers (cell survival?).*

In a few instances, we cultured the explants for 48 hours and observed further shortening or even disappearance of 2^nd^ and 3^rd^row stereocilia in outer hair cells in the presence of the blockers. However, the reviewer is right, the hair cell survival issue does complicate any experiments with prolonged culturing.

*6) One limitation of the study is that experiments were carried out at early postnatal ages. Thus, it is not clear whether similar mechanisms maintain hair bundles in adult mice or only during development. This is a limitation of the study. The authors argue around it but I think this needs to be stated clearly without trying to argue that these are mature stereocilia.*

We do agree that it is important to state clearly all limitations of our study. However, we also think that it is important to provide the arguments in favor of the idea that the observed phenomena may reflect basic mechanisms controlling the height and the shape of transducing stereocilia in mature bundles. Therefore, we re-phrased the relevant part of the Discussion as follows, “An important limitation of our study is that it was performed in young postnatal hair cells with stereocilia bundles that are not entirely mature. […] Therefore, the MET-dependent stereocilia remodeling observed in our study may reflect basic mechanisms controlling the height and the shape of transducing stereocilia after their initial growth.”

*7) One wonders about the physiological relevance of the phenomenon. Are we really looking at a mechanism that is in vivo relevant for maintaining stereocilia length on the long time scale of the experiments? An alternative view could be that the experiments reveal a different underlying physiological process that operates at a much faster time scale. Perhaps increased influx of calcium during active transduction following noise stimulation leads to local changes in actin or its interaction with the membrane that are critical to re-establish tension in the transduction machinery following channel opening thus contribution to adaptation. What is observed here is just an exaggerated manifestation of a different underlying physiological process. May be this could be discussed.*

We have clarified our view on the physiological significance of the observed phenomenon in the Discussion as follows, “What is the physiological significance of the MET-dependent stereocilia remodeling? […] If this is the case, then the MET-dependent remodeling of stereocilia tips may also contribute to processes such as the dynamic regulation of tip link tension”

*Reviewer #2:*

*[…] As mentioned above, the data strongly support the conclusion. However, I am struggling to link these in vitro findings with the hypothetical in vivo scenario. The staircase-like architecture of the stereociliary bundle is mainly acquired during late embryonic and early postnatal stages of development (up to about P4). However, during this critical period the MT current and the MT channel open probability are absent or very small (onset is about P2-P3 in apical OHCs: i.e. Waguespack et al. 2007; Lelli et al. 2009). Therefore, in the in vivo situation it is unlikely that any calcium will flow into the stereocilia during the time when their precise length and width is established. This seems to suggest that calcium is not required for the formation of the stereocilia. Moreover, recent evidence from the Holt group has shown that in the Tmc1/Tmc2 double KO, in which the MT current is completely absent, the stereocilia seem normal, at least in the apical coil. I wonder whether the authors have any thoughts on how their data can be reconciled with the hypothetical in vivo scenario.*

We have added to the Discussion a paragraph on the potential physiological significance of the observed phenomena. “What is the physiological significance of the MET-dependent stereocilia remodeling? […] If this is the case, then the MET-dependent remodeling of stereocilia tips may also contribute to processes such as the dynamic regulation of tip link tension”

*Subsection “Blockage of the MET channels leads to length dysregulation and overall shortening of transducing stereocilia”, second paragraph. Some clarification is required about the growth of stereocilia. The authors have indicated that the taller stereocilia reach their final length at about P4. This is in contrast with previous publications suggesting that final length is reached after P12 in both IHCs and OHCs (e.g. Kaltenbach et al. 1994; Peng et al. 2009).*

We would like to point out that both referenced papers (Kaltenbach et al. 1994; Peng et al. 2009) describe the continuous growth of the tallest row stereocilia throughout P12 and older ages in the *inner* but not *outer* hair cells. In fact, Figure 7 in Kaltenbach et al. (J. Comp. Neurol., 1994) shows clearly that stereocilia bundles of the *outer* hair cells reach their mature height at P2 in the base-middle of the cochlea, at P4 in the apical turn, and at P6 in the very apex. Our experiments were performed in the middle of the cochlea. We have clarified this point in the Discussion as follows, “[…]the developmental growth of stereocilia bundles in the mid-cochlear OHCs is already at the plateau at P4 {Kaltenbach, Falzarano et al. 1994}, while the growth of IHC stereocilia bundles continues until ~P18 {Kaltenbach, Falzarano et al. 1994; Peng, Belyantseva et al. 2009}”.

*Discussion, third paragraph. The information that the observed remodelling is affected by temperature is interesting and it seems to me to be worth showing.*

We apologize for a somewhat schematic description of the much more complex phenomena that we observed at room temperature. We now describe it as, “It is worth mentioning that, despite the remarkable stability of stereocilia, the processes underlying this stability are physiologically vulnerable. Culturing organ of Corti explants at a room temperature of 25°C results in a less prominent MET-dependent stereocilia remodeling and initiates disruption of the hair bundle morphology (data not shown)”. Obviously, these complex temperature effects require a special investigation. However, they are not surprising, having in mind that the height of a transducing stereocilium may be set by a delicate equilibrium between the modulatory influence of the Ca^2+^ influx on actin assembly/disassembly and the rate of delivery of the essential molecules to the stereocilia tips (see last paragraph of the Discussion).

*Reviewer #3:*

*This is a fascinating and exciting paper that provides firm evidence that functional mechano-electrical transducer (MET) channels are required to maintain the height of the 2nd and 3rd rows of the actin-filled stereocilia that comprise the hair bundles of cochlear hair bundles. The work has been performed to an exemplary standard and is, for the most part, very convincing. Whether or not this is 'the first experimental evidence for the role of the MET current in the maintenance of hair bundle structural stability in mammalian hair cells' is somewhat debatable, unless one decides that all data obtained from transgenic mice harbouring temporally conditional alleles do not count as experimental evidence. What this paper does show though, and this is novel, is that the shortening of the rows of the transduction competent stereocilia induced by loss of MET channel activity is rapid (and therefore more likely to be a direct consequence of channel block), entirely reversible, and dependent on calcium influx.*

We do hesitate to consider any previously reported genetic models as a direct evidence for MET-dependent stereocilia remodeling because none of them is proven to disrupt mechanotransduction without any other potential effects on the structural proteins in the stereocilium.

*There are a few points that the authors may want to consider:*

*Abstract and Introduction, second paragraph: Whilst Ca^2+^ influx through the MET channels open at rest may be considered as 'constitutive' there are also active ATPases that pump the Ca^2+^ out so is this influx really the same as Ca^2+^ loading?*

We have changed these sentences to avoid the term “Ca^2+^ loading”.

*Introduction, first paragraph: The hair cells in the avian basilar papilla, for example, do not regenerate or turnover unless the hearing organ is damaged and therefore have to maintain their precisely arranged stereocilia for life.*

We have changed the wording to remove any logical opposition of mammalian and non-mammalian species.

*Subsection “Blockage of the MET channels leads to length dysregulation and overall shortening of transducing stereocilia”, first paragraph: Do MET channels remain functional after long-term (24h) blockage with 100 μm amiloride?*

We have tested MET function after long-term (24h) blockage only for benzamil, because it was important for our “regrow” experiment on Figure 3. However, we see no reasons to believe that other blockers would behave differently.

*Subsection “Blockage of the MET channels leads to length dysregulation and overall shortening of transducing stereocilia”, third paragraph: Is there an explanation for why the largest effects were observed in the 3rd row of OHCs?*

No. We are also puzzled by that.

*Figure 1: Labelling of the y-axis: It is not formally shown that these are transducing stereocilia so it may be more correct to refer to these (as in the text) as 2nd and 3rd row stereocilia.*

Corrected.

*Figure 2: Is the control IHC bundle representative? This bundle has up to 5 rows of stereocilia, the amiloride treated has 4, whilst the benzamil treated has mostly 3. Are the very short rows disappearing too and are these therefore also transduction competent?*

We are very thankful to the reviewer for this question. We quantified the number of supernumerary (4^th^ and 5^th^row and unranked) stereocilia in these experiments. We found that, indeed, MET blockage results in the accelerated “pruning” of these stereocilia in both IHCs and OHCs. For the series shown in Figure 2, the IHCs cultured in control conditions had 25.1 ± 0.9 supernumerary stereocilia per cell (n=31) while those cultured with MET blockers had significantly lower numbers (amiloride = 16.8 ± 1.2, n=17, *P*<0.0001; benzamil = 20.4 ± 1.3, n=19, *P*<0.04). The accelerated pruning of supernumerary stereocilia after MET blockage was also observed after only 5 hours of MET blockage. These results may indicate that the supernumerary stereocilia express functional MET channels. Alternatively, the developmentally-regulated program of retraction of these supernumerary stereocilia may depend on the intracellular Ca^2+^ concentration, which is expected to decrease after MET channel blockage. We have included this finding in the manuscript, see subsection “Blockage of the MET channels leads to length dysregulation and overall shortening of transducing stereocilia”, fourth paragraph.

*Figure 4: It may be useful to define, as shown later in Figure 6, exactly how the height reduction is measured. Presumably the stereocilia with pointed tips indicated by the arrowheads in the benzamil and tubocurarine treated cells are not shortened? Also, the tubocurarine treated hair bundle looks exceptionally hairy. Was this observed in all samples that had been exposed to dTC?*

The explanatory cartoon in Figure 4 has been changed as requested. The reviewer is right, we have measured the distances to the very tips. As to the “hairy” appearance of OHC stereocilia after tubocurarine, this is unlikely to be an effect of the drug – similar “hairy” stereocilia can be observed in some control samples, see for example Figure 6—figure supplement 1, panel A. Generally, “hairiness” increases with the decrease of extracellular Ca^2+^ concentration but the exact reason for sample-to-sample variability is still unknown.

*Figure 5: The actin filaments and plasma membranes are poorly resolved. Was this image obtained from the milled samples? And were the images in Figure 4 obtained using conventional TEM? Please specify.*

While the manuscript has been reviewed, we have obtained better quality images of IHCs. Therefore, we replaced the images in Figure 5. Now, all TEM images in the figures of the manuscript were obtained in the milled samples by FIB-SEM. We have corrected the Methods section, correspondingly.

Subsection “Ca^2+^ influx through the MET channels controls the remodeling of transducing stereocilia”, first paragraph: To save the reader having to read the Peng et al. Neuron paper please state whether BAPTA-AM affects the open probability of the MET channel at rest.

Done.

*Figure 7 and elsewhere: How is a tip link defined? There are often multiple links present at the tips of the stereocilia shown. Are these all counted as tip links irrespective of the direction they are pointing? The 7h post BAPTA treated bundle shown in the lower right hand panel of part D has remarkably few tip links and does not look representative of the data shown in the bar graph below.*

We have added the following paragraph to the Methods section, “Tip link count.A “tip” link was defined as a link that extends obliquely from the top of a lower row stereocilium to the side of a taller stereocilium in the direction of mechanosensitivity of the bundle. Any other link originating at the hemisphere of the tip of a shorter stereocilium was not considered as a tip link. See {Indzhykulian, Stepanyan et al. 2013} for more details”

*Figure 4—figure supplement 1: This is a very pleasing and informative figure.*

Thank you for a positive comment.

*Subsection “Ca^2+^ influx through the MET channels controls the remodeling of transducing stereocilia”, last paragraph: Are long term (up to 1h) changes in the extracellular calcium around cochlear OHC hair bundles likely to occur* in vivo *and if so under what conditions? This may be something that could be discussed or mentioned briefly at this point.*

We are not aware of any physiological conditions that would result in the long-term changes of Ca^2+^ concentration in the endolymph. There is a recent report from the Fridberger group (PD 48, ARO 2017) indicating the possibility that extracellular Ca^2+^ may be concentrated in the tectorial membrane area overlying hair cell bundles. However, the physiological role of these potential extracellular Ca^2+^ sink is not clear.

*Subsection “Stereocilia remodeling after disruption of the tip links”, first paragraph: Is there an explanation for this? Are the MET blockers simply more efficient than extracellular BAPTA? Or does this mean that stereocilia can shorten without first getting thinner at their tips?*

Yes. This is an expected result due to the complete elimination of the MET current after tip link disruption while MET blockers don’t eliminate the MET current completely since we used their non-saturating concentrations. We clarified our idea as follows, “This is expected because tip link disruption with BAPTA eliminates the MET current completely, in contrast to the experiments with non-saturating concentrations of MET blockers that may result in a Ca^2+^ gradient across the stereocilium diameter and the preferential remodeling of peripheral actin filaments (Figure 6)”.

*Subsection “Stereocilia remodeling after disruption of the tip links”, last paragraph: Would these re-analysed data also be consistent with some element of tension being provided by the tips links climbing up the stereocilium?*

Yes, our data are consistent with the tension provided by the tip link as previously proposed. However, the mechanistic link between tension and stereocilia remodeling has been unknown. Our data suggest this link – the tip link tension generates Ca^2+^ influx through the MET channels, which in turn initiates stereocilia remodeling. Of course, we cannot also exclude the direct effect of mechanical tension on the stereocilia actin remodeling.

*Figure 4 onward use a measurement of stereocilia steps as opposed to heights. Is there a reason for this?*

Yes, we clarified the reason in the Figure 4 legend as follows, “Note that the staircase “step” measurement procedure requires fewer calculations than the estimation of the absolute height of the stereocilium (as in Figure 1and Figure 2) and, therefore, it is more accurate for quantifying smaller changes in the staircase morphology of the bundles (see Figure 1—figure supplement 3).” Since all other experiments explore relatively subtle effects, we used step measurements.

*The Discussion is rather long and very speculative. Many possible mechanisms and molecules are suggested to be involved none of which have been tested. Surprisingly there is no discussion of what determines the final height of the 2nd and 3rd rows when they recover from transient MET channel block.*

We did consider different versions of the Discussion. Unfortunately, the only reasonable approach to discussing potential mechanisms of MET-dependent stereocilia remodeling was either all or nothing, since we cannot yet choose one or two most likely hypotheses from the list of possibilities currently described in the Discussion section. To answer the second part of this comment, we expand the Discussion as follows, “By setting a certain tension of the tip link and MET current at rest, the myosin motors may set this equilibrium at a precise stereocilium height and determine, for example, the final heights of the 2^nd^and 3^rd^row stereocilia when they re-grow after the washout of MET blockers (Figure 3).”.

*Discussion, fifth paragraph: Only one of the three papers cited, the Xiong et al. paper describing the Tmhs-/- mouse, appears to provide evidence for changes in the length in the transducing but not the tallest row of stereocilia. The stereocilia in the third row appears mainly to shorten, but the range of stereocilia steps between the first and second rows and second and third rows becomes extremely variable, suggesting some in the 1st and 2nd rows are shrinking and/or others may even be getting longer (see Figure S1 in Xiong et al., 2012; but note that the step size is unlikely to be in mm as stated).*

Yes, the actual measurements of stereocilia steps were performed only in Xiong et al. paper (2012). However, the published images in other papers indicate similar disruptions of the hair bundle morphology. We now reference directly these images. We also re-phrased the sentence as follows, “Furthermore, deletions in the genes encoding currently known components of the MET machinery *–* TMC1/TMC2 (see Supplemental Figure 4 in {Kawashima, Geleoc et al. 2011}), TMHS (see Figure 1 in {Xiong, Grillet et al. 2012}), TMIE (see Figure 5 in {Zhao, Wu et al. 2014}) – result in the loss of MET current and changes of the hair bundle morphology that seem to be limited to the shortest but not tallest row stereocilia in the auditory hair cells”. As to the potential difference between the effects of MET channel blockers and stereocilia phenotype in mutant mice, we already mentioned that none of these mutant models has been proven to disrupt mechanotransduction without any other potential effects on the structural proteins in the stereocilium.

Subsection “Scanning Electron Microscopy (SEM)”: graded not grated; delete 'of' from '5 nm of platinum'.

Corrected (subsection “Scanning Electron Microscopy (SEM)”).